# Reinforcing materials modelling by encoding the structures of defects in crystalline solids into distortion scores

Alexandra M. Goryaeva [1✉], Clovis Lapointe[1], Chendi Dai[1], Julien Dérès[1], Jean-Bernard Maillet[2] & Mihai-Cosmin Marinica [1✉]

This work revises the concept of defects in crystalline solids and proposes a universal strategy for their characterization at the atomic scale using outlier detection based on statistical distances. The proposed strategy provides a generic measure that describes the distortion score of local atomic environments. This score facilitates automatic defect localization and enables a stratified description of defects, which allows to distinguish the zones with different levels of distortion within the structure. This work proposes applications for advanced materials modelling ranging from the surrogate concept for the energy per atom to the relevant information selection for evaluation of energy barriers from the mean force. Moreover, this concept can serve for design of robust interatomic machine learning potentials and high-throughput analysis of their databases. The proposed definition of defects opens up many perspectives for materials design and characterization, promoting thereby the development of novel techniques in materials science.

[1] Université Paris-Saclay, CEA, Service de Recherches de Métallurgie Physique, Gif-sur-Yvette 91191, France. [2] CEA - DAM, DIF, Arpajon Cedex F-91297, France. ✉email: alex.goryaeva@gmail.com; mihai-cosmin.marinica@cea.fr

A perfect crystal is a purely theoretical concept. Real-world crystals contain imperfections, also called defects. Some simple defects, such as vacancies, are always present in crystals at a concentration of thermodynamic equilibrium. The concentration and morphology of defects influence the properties of crystalline solids. For instance, the scattering of electrons and phonons on defects underlies the electronic and thermal conductivity. Furthermore, the energy and kinetics of defects essentially control the material's plasticity, viscosity and evolution of its microstructure. As a result, the ability of crystalline materials to fulfil a set of design criteria is controlled by static and kinetic properties of defects population, either in thermodynamic equilibrium or non-equilibrium. Identification and characterization of defects provide crucial information for interpretation of simulations and experiments that bridge the gap between atomic and micrometre scales. This work introduces a novel concept of defect characterization at the atomic scale with the aim to reinforce the cutting-edge methods of materials modelling, such as free energy evaluation from the mean force, quantum mechanics/molecular mechanics (QM/MM) simulations and the design of robust interatomic machine learning (ML) potentials.

Present-day materials science enables simulations of defect nucleation, recombination, migration and transition at the atomic scale by means of ultra large scale experiments[1–3]. Facilitated by the continuous increase in computational power and parallel computing, these objectives are achieved using traditional molecular dynamics (MD), quantum-classical QM/MM simulations[4–6] and by a rapidly growing number of fast exploring, biased in energy[7] or mean force[8,9] methods and other simulation schemes, such as accelerated MD[10] or statistical learning approaches[11]. However, the application of these methods is often hindered by the general inability to extract the relevant information about the defects or to define a suitable set of collective variables that drive the physical process. Moreover, an accurate interpretation of these calculations requires processing enormous amounts of data, to select the information related to the defects. Understanding which particles are associated with defects, and which belong to the bulk structure, is not trivial. The vast majority of methods for structural identification are based on geometrical analysis of local atomic environments (LAEs), e.g., coordination analysis, bond-angle and common neighbour analysis[12,13], Voronoi cell and polyhedral template matching[14,15], etc. In order to accurately analyse and identify a defect structure, the geometry-based order parameters should be complemented with some local physical properties. Most commonly, the relevant properties, such as energy or stress per atom[3,16], are derived from a series of force field calculations. However, these properties are not always available, which hampers a universal strategy of structural analysis. For instance, energy and stress per atom cannot be directly extracted from the widely used ab initio plane-wave (PW) methods. In this case, a post treatment, such as projection on local orbitals or Mulliken analysis, is needed. In some multiscale simulations, e.g., in QM/MM, even the concept of total energy is not well defined. Thus, introducing a defect detection strategy that is (i) independent of the force field method and, (ii) at the same time, can quantitatively describe the distortion degree of each atomic environment, will improve the means and universality of defect characterization. Here, we propose a method based on the so-called distortion score of atomic environments, which can be naturally provided by the distance-based ML outlier/anomaly detection methods.

Detection of deviating instances is of primary importance in many disciplines, such as economics and finances[17,18], medical diagnostics and image processing[19–21], psychology and social sciences[22,23], meteorology and climatology[24,25], etc. The practical importance of outlier and novelty detection has led to the development of multiple numerical approaches, based on robust statistics[26,27], support vector machine (SVM) methods[28,29], neural networks (NNs)[30,31], Bayesian formalism[32,33], etc. For the majority of these methods, the outlier detection task is solved in a feature space by distinguishing the normal data instances (inliers) from other data points. The description of inliers is learned by constructing a model with well sampled data instances. The unseen samples are then compared to the learned data patterns and characterized by a score or distance, which describes the proximity of new instances to the inliers. This distance is compared to a decision threshold of the trained model and the tested data are classified as outlier if the critical threshold is exceeded. In materials science, outlier detection methods are still rarely applied for atomic systems and rather serve as a preliminary step, needed to isolate the perfect structure[34].

In the present study, we propose to use the distances provided by outlier detection models, such as minimum covariance determinant (MCD) or support vector machine (SVM) methods, as a quantitative description of LAEs, hereafter called distortion score. Based on these local distortion scores, we identify structural defects as atoms-outliers deviating from the bulk structure. This strategy is well adapted for detection of structural defects and monitoring their trajectories, as well as for tracking the structural changes during phase transitions or crystallization. We demonstrate how the stratified definition of defects based on the local distortion scores can serve for reconstruction of energy profiles in mean force calculations. Furthermore, the defect detection is coupled with ML techniques to establish a qualitative criterion for transferability/reliability of kernel ML potentials for modelling a given defect structure.

## Results

**Distortion score and its correlation with energy per atom**. The distortion score of LAEs describes a statistical distance from a reference distribution in the feature space of atomic descriptors, such as those described in refs. [35,36]. The reference distribution can be constructed from LAEs of a defect-free crystalline system at a given temperature or from a subset of atoms of particular interest. Figure 1a depicts the schema for computing the distortion score with respect to defect-free bulk structure. The training data set is formed by reference LAEs of the bulk structure represented in the feature space of atomic descriptors. The reference distribution is then learned by a ML algorithm. In this study, we mainly use the MCD[27,37]. To the best of our knowledge, MCD has never been applied for the needs of atomistic materials science. MCD is an affine equivariant estimator, i.e., the data might be rotated, translated or rescaled (e.g., due to a change of the measurement units) without affecting the results[27]. It is worth mentioning that MCD is tailor-made for unimodal distributions. Consequently, a careful selection of the training data should be performed (see Supplementary Note 2 for more details).

The distortion score is computed for each atom in the analysed system via computing the statistical distance of the LAE with respect the learned distribution of the reference structure LAEs. The distortion score from MCD corresponds to the robust distance $d_{RB}$ (see "Methods", Eq. (5)). Figure 1b shows the distortion scores computed for a simulation cell with 132 atoms, which contains four self-interstitial atoms forming a three-dimensional (3D) C15 cluster[38] in bcc Fe. The detected cluster of atoms outliers (Fig. 1b, inset A) includes the defect itself and its nearest atomic environment. The difference in magnitude of the distortion scores within the outlier cluster enables the stratified description of the defect and allows to distinguish the zones with different level of atomic distortion (as depicted with dashed grey lines in Fig. 1b). The atoms forming the defect (Fig. 1b, inset C) are

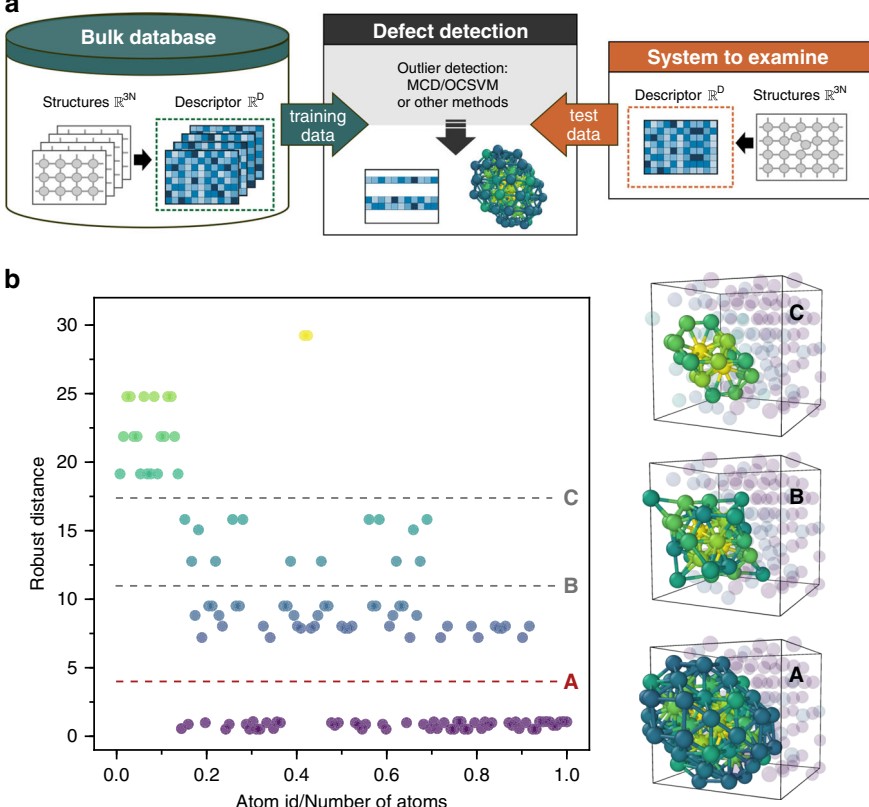

**Fig. 1 Defect detection and stratification based on the distortion score. a** Scheme of the defect detection. The training data set consists of the defect-free bulk structures. The structures for the training and test are represented in the same feature space $\mathbb{R}^D$ of atomic descriptors. To perform defect detection, each atomic environment from the test system is compared to the learned bulk structures and characterized by distortion score. Atoms with scores above the critical threshold are classified as structural outliers that form the defect. **b** Detection and stratification of four self-interstitials cluster with C15 morphology[38]—$I_4^{C15}$—based on the distortion score provided by robust MCD. Each point on the plot represents an atom in the simulation box. The colour of points corresponds to the colour of atoms in the inset structures. The threshold between the bulk (atoms-inliers) and defect (atoms-outliers) is indicated with a dashed red line, labelled as A. The grey dashed lines B and C indicate the possibilities for defect stratification. The defect structures A, B, C are obtained using the corresponding thresholds.

characterized by bigger $d_{RB}$ distances compared to their nearest environment. Here we exemplified the case with single type of reference structure, given by the bcc bulk. Each LAE can be characterized by a multi-dimensional distortion score, subsequently computed with respect to various reference structures, e.g., to different structural types of bulk or even to the structures of particular defects of interest (see the analysis of a displacement cascade in Supplementary Note 2).

When computed with respect to the distribution of the underlying bulk structure, the distortion score exhibits a correlation with the local atomic energy (Fig. 2). Both concepts, local atomic energy and distortion score, encode the local geometric information. The link between the local atomic energy and the LAEs was established in the early days of atomistic materials science. For metals, the tight binding approximation[39,40] has formalized the basis of this relation.

With the appearance of semi-empirical potentials[40–42], the tight binding second moment was replaced by ad-hoc local functions that should be fitted against the bulk properties, defect formation and migration energies, etc. Not limited to metals, the functional form of the local energy on the local coordination is the basis of empirical many-body force fields. These functions have simple analytic forms, such as the number of first and/or second neighbours, radial functions[43–45] or somewhat more complex functions accounting for angular information[46]. Regardless of the analytic form, all these functions have the same utility and provide the fingerprints of

atomic environments. Furthermore, the present-day ML potentials[47–49] propose a direct multivariate regression, in the descriptor space, between the LAE and the atomic energy. Here we demonstrate that the geometric information of LAE, encoded via MCD robust distance $d_{RB}$, is intrinsically related to the local atomic energy (see the "Methods" section). Figure 2 reports the observed correlation between the distortion score $d_{RB}$ and the local atomic energy in bcc Fe. The comparison is performed for the atomic arrays with three classes of structural defects: vacancies, self-interstitials and stacking faults (SFs; also called γ-surfaces). These configurations are included in the training database of the Gaussian Approximation Potential (GAP) for Fe[50]. The atomic energies were computed using the same potential. The kernel formalism of GAP potential ensures the high accuracy of the atomic energy of the training configurations[50]. For all three defect classes (Fig. 2), the determination-correlation coefficient $R^2$ between $d_{RB}$ and local energy is higher than 80%. The present approach completes the previous observation of Sharp et al.[2] in grain boundaries. The study[2] monitors the likelihood of atoms to rearrange within the grain boundaries through the so-called softness of atoms. The softness is a continuous, signed, scalar quantity that captures the relevant properties of the LAEs based on the binary classification using SVM. Likewise, the potential energy of atom is positively correlated with its softness[2], although there is a large spread for a given energy value. In this study, we observe the higher variance of $d_{SVM}$ compared to the statistical distances $d_{RB}$,

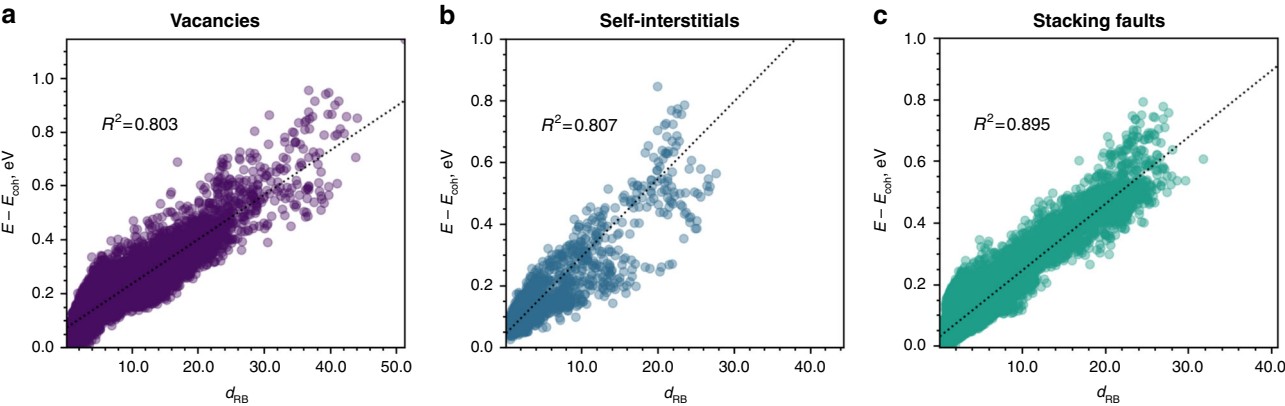

**Fig. 2 The correlation between energy per atom and the distortion score.** The distortion score is described via robust MCD distance $d_{RB}$ in bcc Fe systems with: **a** vacancies; **b** self-interstitials; **c** stacking faults. Each point on the plot represents an individual atom in a simulation box. The atomic arrays are taken from the GAP potential database[50]. The correlation is performed over 103,000 LAEs and each defect class gathers diverse instances from 0 K static relaxation to molecular dynamics simulations at various temperatures. MCD analysis is performed on the structural data represented using bispectrum SO(4)[36,48] with the angular moment $j_{max} = 4.5$. The atomic energies are computed with the GAP potential[50].

consistent with that previously reported by Sharp et al.[2] (see Supplementary Note 1).

The remarkable accuracy in the relation between the distortion score described via statistical distances and the local energy (see the "Methods" section) opens up many perspectives for further developments in analysis and modelling of defects in crystalline solids. To demonstrate the importance and perspectives of the present concept, we present in the following sections three promising applications of the stratified definition of the defects.

**Application 1: detection and structural analysis of defects.** Based on topology, defects are generally classified as 0D or point defects, one-dimensional (1D) or line defects, two-dimensional (2D) or planar defects, and 3D defects. Structural analysis of different defect classes typically requires using different strategies of structural analysis[14,15,51], which impends a universal strategy for defect identification. Here we propose a universal scheme for localization and analysis of defects based on the distortion score provided by robust MCD and consider the examples of cubic metals, fcc Al and bcc Fe (Fig. 1).

The conventional geometry-based techniques for structural analysis are often sensitive to atomic perturbations[14,52]. This shortcoming may hamper structural interpretation in systems at high temperature and/or under large deformation. Here, to avoid sensitivity of the defect detection model to atomic perturbations, the defect-free training data set incorporates systems with some noise around the perfect atomic positions (see "Methods" for more details). In this section, the structural data are represented in the feature space of bispectrum SO(4)[36,48]. This type of atomic descriptor was previously used for the development of ML interatomic potentials[47–49].

In Al, the outlier detection strategy was tested for the typical defects for fcc structures, namely for the mobile $\frac{1}{2}\langle 110\rangle\{111\}$ loop, the sessile $\frac{1}{3}\langle 111\rangle\{111\}$ Frank loop and for the $\frac{1}{2}\langle 110\rangle\{111\}$ edge dislocation. All the defect structures are correctly identified based on the distortion score metrics (Fig. 3). In contrast to the $\frac{1}{2}\langle 110\rangle\{111\}$ loop (Fig. 3b), the $\frac{1}{3}\langle 111\rangle\{111\}$ Frank loop (Fig. 3a) contains a SF, which prevents it from gliding. In fcc structures, $\frac{1}{2}\langle 110\rangle$ dislocations dissociate into two dislocation partials separated by the SF according to the reaction $\frac{1}{2}\langle 110\rangle \rightarrow \frac{1}{6}\langle 211\rangle + \frac{1}{6}\langle 21\bar{1}\rangle$. The dissociated dislocation core described via the distortion score (Fig. 3a) is compared with those from the energy per atom calculations and from the common-neighbour analysis (CNA). The three methods are consistent in identification of the dislocation partials $\xi_1$ and $\xi_2$ (Fig. 3c). However, structural analysis based on the distortion score better reproduces the core spreading than CNA. The CNA analysis identifies a structural type of each atomic environment without providing any appropriate measure of distortion within a given structural class, which hampers estimation of the core spreading with this method.

For bcc Fe, we examine the performance of outlier detection methods for point defects and their clusters (Fig. 4a–d), SFs (Fig. 4e) and $\frac{1}{2}\langle 111\rangle$ screw dislocation dipole (Fig. 4f). It is worth emphasizing that the structures of Gao-triangles $I_2^{NP}$, also called non-parallel clusters, and $I_4^{C15}$ self-interstitial atom (SIA) clusters (Fig. 4c, d) are often misinterpreted by conventional geometry-based methods. The Gao-triangle configuration $I_2^{NP}$ (Fig. 4c) is a SIA cluster with three interstitial atoms in the {111} plane and one vacancy in the in centre of triangle. This interstitial defect is the precursors of the C15 Laves phase clusters[38]. The C15 cluster $I_4^{C15}$ (Fig. 4d) has a well-defined 3D crystallographic structure being close to two attached Frank–Kasper polyhedra. Both $I_2^{NP}$ and $I_4^{C15}$ are immobile and very stable and, therefore, they represent important instances in the energy landscape of SIAs in bcc Fe[38]. Due to their structure, the $I_2^{NP}$ and $I_4^{C15}$ defects can be only partially detected by the Wigner–Seitz analysis and require the use of complementary methods, such as polyhedral template matching (PTM)[15] or energy per atom calculations. The tested robust MCD approach exhibits an excellent performance for these complex defect structures and, in contrast to the conventional methods, it implies neither preliminary knowledge of the defect structure (for effective PTM) nor energy per atom calculations. This is especially valuable for the detection and the characterization of previously unseen defects, which, for instance, can form in materials under extreme conditions.

**Application 2: distortion score for mean force calculations.** The proposed stratified definition of defects can be of great help for calculations where relevant local properties from interatomic force field are not available. For instance, in the case of widely used PW electronic structure calculations, the definition of energy per atom is ambiguous and requires to project delocalized electron density on local atomic orbitals.

The definition of energy profile is critical in many statistical learning approaches, including QM/MM methods, which are currently at the forefront of computational materials science[4–6].

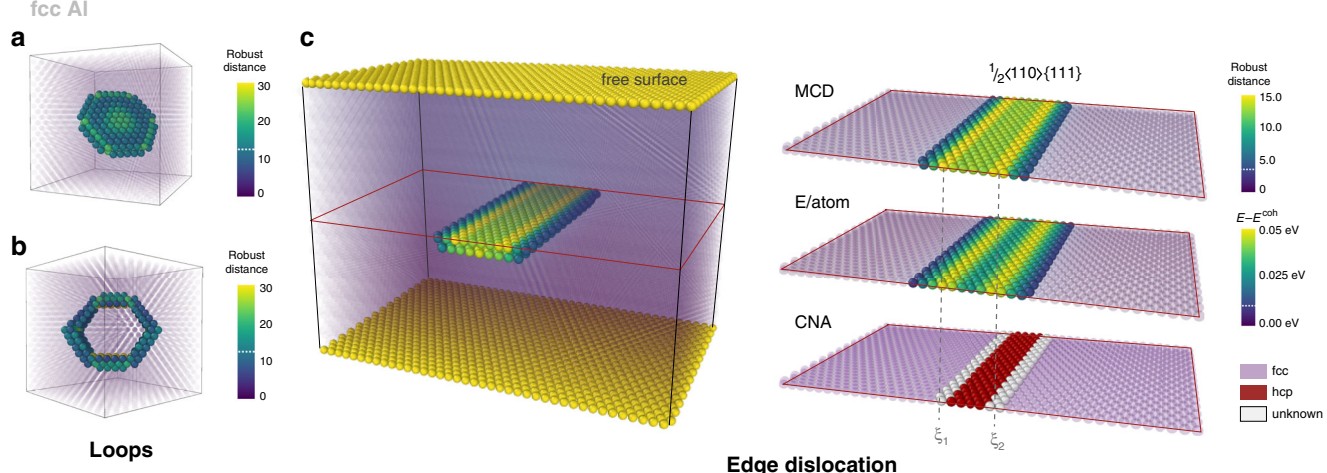

**Fig. 3 Structural defects in fcc Al detected using the distortion score. a** Frank loop $\frac{1}{3}\langle 111\rangle\{111\}$; **b** $\frac{1}{2}\langle 110\rangle\{111\}$ loop; **c** $\frac{1}{2}\langle 110\rangle$ dissociated edge dislocation. The atoms are coloured according to their distortion score, as provided by robust MCD distance. The atoms identified as fcc bulk are shown in transparent purple. The dissociated edge dislocation core structure **c** is compared with those provided by energy per atom (E/atom) and CNA. The energy per atom is calculated using EAM potential by Liu et al.[69]. The two dislocation partials are indicated as $\xi_1$ and $\xi_2$.

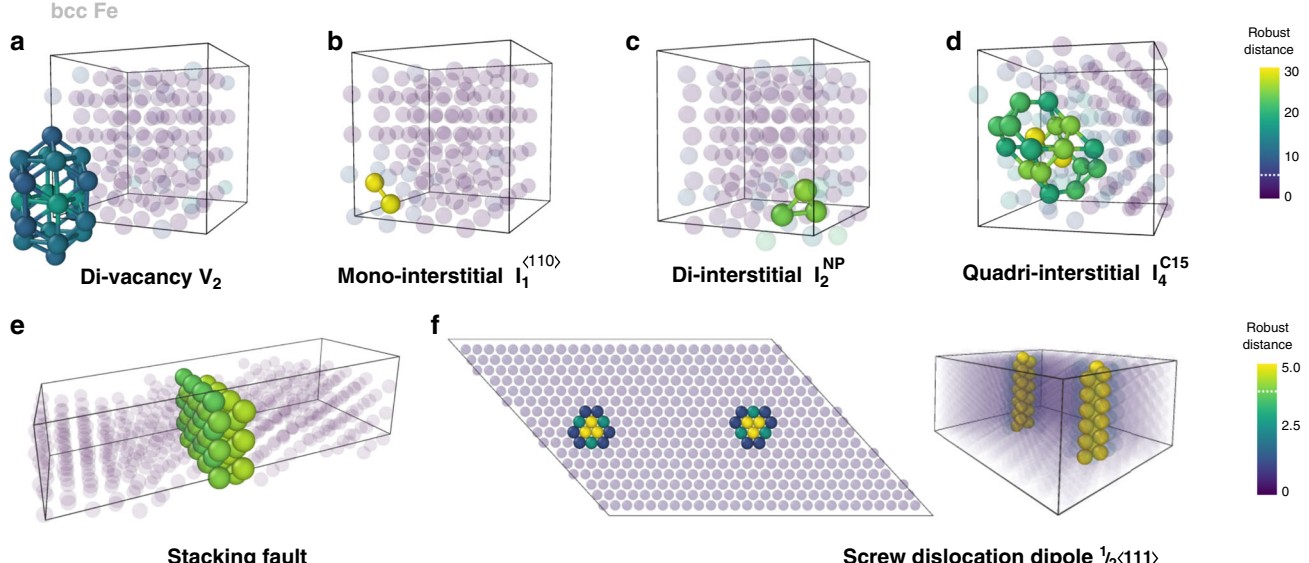

**Fig. 4 Structural defects in bcc Fe detected using the distortion score. a** Divacancy; **b–d** various interstitial defects; **e** stacking fault; and **f** $\frac{1}{2}\langle 111\rangle$ screw dislocation dipole. The types of defects are indicated on each subplot. The atoms are coloured according to their distortion score, as provided by robust MCD distance. The atoms identified as bcc bulk are shown in transparent purple. For the interstitial clusters (**b–d**), the atoms with the robust distance $d_{RB} < 17$ are set transparent.

In this method, the system commonly consists of the two parts: the core, which is described using ab initio, and the outer part, which follows classical mechanics or surrogate tight binding Hamiltonian (the main contribution that has fast force evaluation). The interaction of these parts and description of the whole system are given solely by the forces, which are well defined local quantities. However, the total energy of the system cannot be well defined in this case. Moreover the wavefunction of the core part is highly perturbed by the buffer region between the two parts of the system, which makes the attempts to define the local energy difficult. As a consequence, QM/MM methods cannot have access neither to local nor to total energies.

Without direct access to the energy of the system, the migration and transformation energy barriers can be fully recovered from the atomic forces using the mean force concept[8,9] both for the 0 K[53] and finite temperature calculations[10]. Here we consider an example

of $P$ images from a migration trajectory obtained using a standard pathway method, e.g., nudged elastic band (NEB)[54]. In this migration path, $\mathbf{q}_i \in \mathbb{R}^{3N}$ is the $i^{th}$ image along the system trajectory. The path is indexed by a reaction coordinate $\zeta \in [0, 1]$ in such a way that $\mathbf{q}(\zeta = 0) = \mathbf{q}_1$ and $\mathbf{q}(\zeta = 1) = \mathbf{q}_P$. This reaction coordinate can be achieved by a spline interpolation of all the intermediate NEB images along the migration pathway. The corresponding energy profile can be then recovered from the mean force $\partial_\zeta F(\zeta)$[8,9], i.e., the derivative of the free energy $F(\zeta)$ with respect the reaction coordinate:

$$\Delta E(\zeta) = E(\zeta) - E(0) = \int_0^\zeta \partial_{\zeta'} F(\zeta')d\zeta'. \quad (1)$$

The above equation is the exact form of the 0 K energy profile along the migration pathway that can effectively circumvent

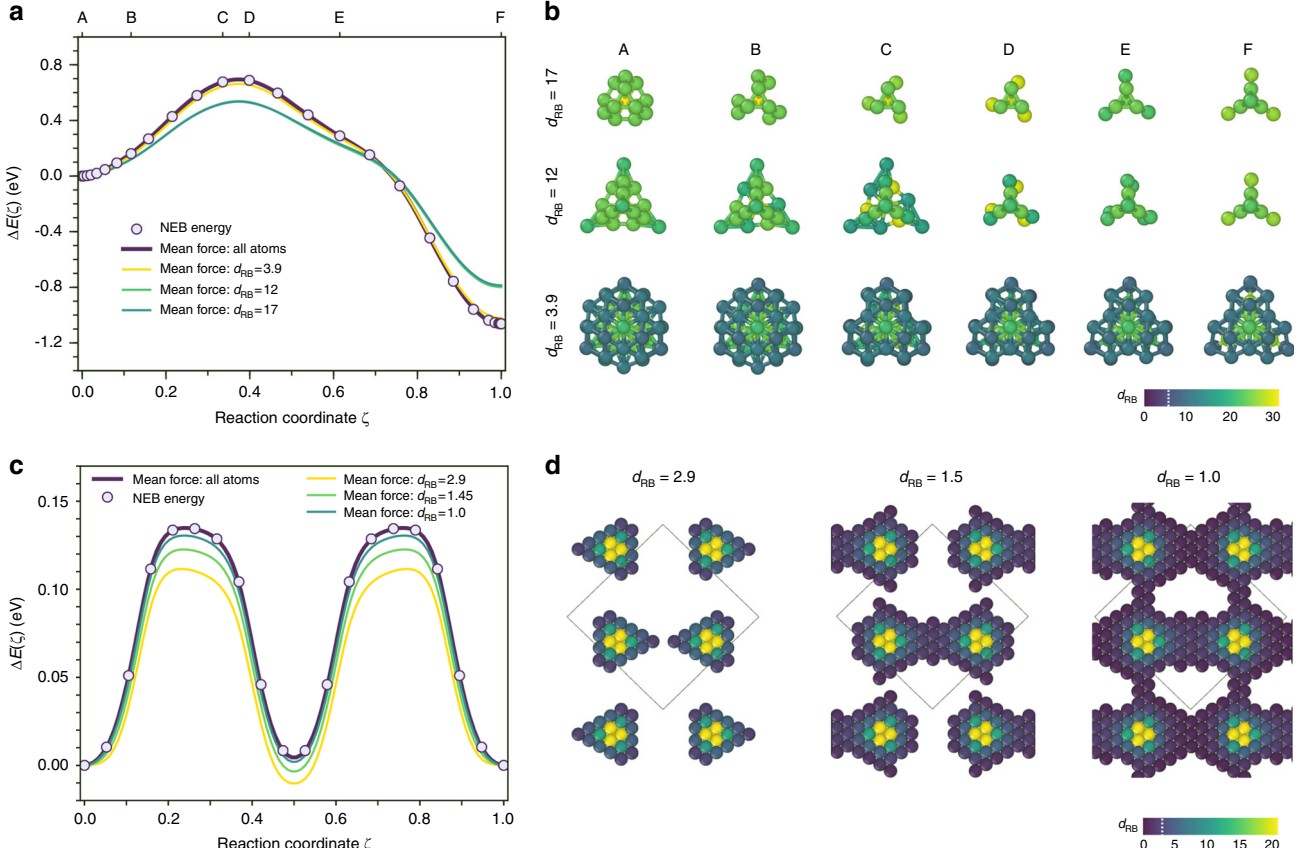

**Fig. 5 Reconstruction of defect energy profiles from mean force calculations. a** Energy profile along the $I_2^{C15} \rightarrow I_2^{NP}$ SIA transition pathway in bcc Fe. Comparison of the total energy NEB calculations with the mean force integration over the confidence region $\nu_{MCD}$ defined by different distortion score cutoff. The calculations are performed using Marinica EAM potential[38,45]. **b** Evolution of the defect structure along the $I_2^{C15} \rightarrow I_2^{NP}$ transition path provided by the distortion scores based on the robust MCD analysis. The stratified defect structures are shown for the cutoff distortion scores $d_{RB} = 3.9$ (full defect cluster), $d_{RB} = 12$ and $d_{RB} = 17$. The atoms are coloured according to their distortion scores. The colour code corresponds to the scale bar provided in **d**. The depicted atomic clusters are oriented along the $\langle 111 \rangle$ direction. **c** Energy profile of the screw $\frac{1}{2}\langle 111 \rangle$ dislocation dipole glide in {110} plane in bcc Fe. Comparison of the total energy NEB calculations with the mean force integration over the confidence region $\nu_{MCD}$ defined by different distortion score cutoffs. The calculations are performed using Ackland-Mendelev EAM potential[44]. **d** The stratified structures of the dislocation cores with different distortion score $d_{RB}$ based on the robust MCD analysis. The critical MCD threshold of the bulk structure is $d_{RB} = 2.9$. The atoms are coloured according to their distortion scores. The depicted structures are oriented along the $\langle 111 \rangle$ direction.

direct total energy calculations along the pathway. Using the explicit form of the mean force $\partial_\zeta F(\zeta)$ and derivatives of the spline interpolation of atomic coordinates[10,53], the migration energy profile becomes:

$$\Delta E(\zeta) = -\sum_{i\in box}\sum_{\alpha=x,y,z}\int_0^\zeta \frac{\partial q_{i\alpha}(\zeta')}{\partial\zeta'}f_{i\alpha}(\zeta')d\zeta', \qquad (2)$$

where $f_{i\alpha}$ is the force acting of the $i^{th}$ atom along the Cartesian $\alpha = x, y$ or $z$ direction; $q_{i\alpha}(\zeta')$ is the interpolated coordinate of the same atom with the $\zeta'$ as reaction coordinate. Figure 5a compares the energy profile obtained directly from NEB calculations with those from the mean force (Eq. (2)) integration. When integrating over the forces of all atoms in the system, the agreement between the two energy barriers is excellent (Fig. 5a).

However, in calculations like QM/MM, it is impossible to take the forces on all atoms. As such, a confidence region with major contribution to the mean force of the system should be defined. As a possible solution, a geometrical cutoff around the defect can be applied[53]. This simple approach is sufficient for the calculations of particular class of compact defects, like interstitial clusters, but it does not provide a universal solution, e.g., it is not applicable for the defect structures that cannot be well localized,

like dislocations. Here we suggest using the distortion score to define the confidence region based solely on geometric information of LAEs. The atoms from the core and the outer part of the system are treated on the same footing. Using the distortion score as local information we are able to indicate the atoms that are more likely to contribute to the mean force of the system. Finally, we integrate the mean force along the complex reaction coordinate and find the migration/transformation energy barrier for systems where the energy cannot be directly defined. For such a defect cluster, the expression of the energy profile becomes:

$$\Delta E(\zeta) \sim -\sum_{i\in\nu_{MCD}}\sum_{\alpha=x,y,z}\int_0^\zeta \frac{\partial q_{i\alpha}(\zeta')}{\partial\zeta'}f_{i\alpha}(\zeta')d\zeta', \qquad (3)$$

where $\nu_{MCD}$ is the confidence region defined by the set of atoms with $d_{RB}$ bigger than a critical threshold. The geometric criterion in direct Cartesian space is replaced here by the distortion score of LAEs. The energy barriers obtained from the mean force integration (Eq. (3)) of atomic clusters and screw dislocations in bcc Fe with different $d_{RB}$ cutoff are reported in Fig. 5. Figure 5a depicts the minimum energy pathway of the $I_2^{C15} \rightarrow I_2^{NP}$ transformation. For these defects, all the atoms with $d_{RB} > 3.9$ are identified as structural outliers by robust MCD (Fig. 1b). The

number of atoms in the detected defect clusters (Fig. 5b, $d_{RB} = 3.9$) varies from 57 to 32 along the transition path. The mean force integration of these clusters is in a good agreement with the reference NEB curve. When increasing the cutoff distance $d_{RB}$ up to 12 and 17 (defect stratification according to Fig. 1b, lines B and C), the nearest environment of the defect is disregarded. This allows to better visualize the transition mechanism (Fig. 5b). However, at the same time, it results in underestimated energy barriers (Fig. 5a). Thus, the contribution of mild outliers into the system's mean force is important and cannot be neglected.

The selection of a confidence region based on distortion score can be especially useful for the reconstruction of the energy profiles in situations where the relevant region is not local and hardly can be grasped using a geometrical cutoff around defects. Figure 5c illustrates the Peierls barrier of a $\frac{1}{2}\langle 111 \rangle$ screw dislocation dipole gliding in {110} plane in bcc Fe. In the depicted simulation cell (Fig. 5d), the dislocations are only distant by 17.45 Å, which imposes a strong elastic interaction between the cores. The complex interaction is deconvoluted using various cutoff of the distortion score $d_{RB}$ (Fig. 5d). The extracted information is subsequently used to reconstruct the migration energy profile. In contrast to the above defects (Fig. 5a, b), the local definition of the dislocation core is not sufficient to accurately reconstruct the Peierls barrier. When considering exclusively the atom outliers (Fig. 5d with $d_{RB} = 2.9$), the barrier is underestimated by more than 20%. Hence, it is necessary to include distorted bulk in the confidence region for the mean force integration. The elastic interaction of dislocations produces relaxation patterns that are captured by the distortion score (Fig. 5d). Including the relevant bulk atoms improves the energy barrier (Fig. 5c). Thus, we are able to reconstruct the NEB barrier within 4 meV deviation, i.e., with more than 95% accuracy. Such analysis and reconstruction of the Peierls barrier also holds for bigger simulation cells (see Supplementary Note 3) with less important interactions between the dislocation cores.

These results open up many perspectives in computational materials science. Beyond the selection of relevant structural information, the detected patterns of atoms can indicate the areas with strong interaction between defects or/and non-homogeneous distribution of strain in the simulation cell. This information is useful in QM/MM to qualitatively verify the convergence of the calculations as well as to handle the frontier between the QM and MM domains. Moreover, the automatic selection of relevant atoms can set the basis for finding appropriate collective variables, which is currently recognized as a critical problem that hinders implementation of free energy methods using automated and unsupervised simulation schemes[8,10].

**Application 3: analysis of kernel ML potentials.** Nowadays, ML force field models represent a worthwhile alternative to conventional interatomic potentials. The vast majority of existing ML force fields for MD calculations are based on kernel methods[11,48,55,56]. Accuracy and numerical cost of these potentials intrinsically depend on the diversity and number of LAEs $M$ in the training database. The force fields built within the GAP framework[48] are among of the most commonly used ones. For the structures close to those from the potential database, GAP can be as accurate as ab initio methods[48,50,57]. However, application of these potentials for modelling configurations beyond the potential database is rarely discussed.

Uncertainty quantification of the Gaussian process regression can provide a qualitative estimate of the potential's accuracy for each atom in a given system. An example of such an estimation was recently demonstrated in ref. [57]. The local error is an appropriate measure of the potential reliability; however, its computational cost ascends to $M^2$, whereas the MD calculations with GAP scale linearly with the size of the database $M$. Here we propose a less costly strategy, able to provide a qualitative estimate of the potential's transferability for modelling targeted defects. The method is based on the outlier analysis and performs examination of defect clusters from the potential database and compares them with the defect structures of interest. Figure 6 illustrates a general workflow for the proposed transferability analysis strategy.

As a study case, we examine the performance of GAP potential for bcc Fe[50]. We have tested this potential to compute various radiation-induced defects, including those beyond the potential database. The results are reported in detail in the Supplementary Note 4. Overall, the GAP potential is remarkably more accurate than any existing semi-empirical potential. However, for few defects, the tested potential exhibits a limited transferability. Among the examined defect structures, we identify (i) the C15 clusters and (ii) the saddle-point configuration $V_3^{max}$ of tri-vacancy migration as "failed" system to test further. For the small size $I_{2,3}^{C15}$ clusters, GAP potential provides the formation energies ca. 2.5 eV higher than those of SIA dumbbells (Supplementary Fig. 10b). This yields an impossible formation of C15 in bcc Fe, which is not consistent with the density functional theory (DFT) predictions[58]. For the tri-vacancies $V_3$, the computed migration energy barrier $V_3^{max}$ is almost 60% lower than the DFT migration energy (Supplementary Fig. 11b). Such an error will have an impact on predictions of defect kinetics under irradiation and interpretation of processes during resistivity recovery experiments[59].

Besides these two defects, we also examine (iii) $\frac{1}{2}\langle 111 \rangle$ screw dislocation core and (iv) its saddle point configuration on the top of the Peierls potential. These structures were not explicitly included into the GAP database; however, the potential performs as accurate as ab initio methods for these defects[60]. The ML algorithm that underlays the GAP potential, Gaussian Processes, is non-parametric and can integrate all the information provided by the projection of the database into the descriptor space $\mathbb{R}^D$. Most likely, the "failed" configurations (i)–(ii) deviate from the defects in the training database, whereas the dislocation structures (iii)–(iv) are similar to those learned by the potential. To check this assumption, we have examined how the defect clusters (i)–(ii) are related to the defect structures from the potential database. For the dislocations (iii)–(iv), we only employ the detected LAEs of SFs as a training data for the transferability analysis. The latter will allow to estimate if accurate modelling of dislocations can be ensured by the presence of SFs in the potential database. The majority of atoms in the "failed" defect clusters (i)–(ii) (Fig. 7a, b) are identified as pronounced outliers, characterized by negative SVM distances. Consequently, the GAP potential mainly performs in extrapolation regime for these defects. The predictions in this regime are not necessarily accurate. Hence, it is not surprising that the energy profiles of those defects predicted by GAP do not agree with DFT calculations. In contrast, the dislocation cores (iii)–(iv) (Fig. 7c, d) do not contain any anomalous instances. Thus, the structural information provided by the SFs was sufficient to ensure good accuracy of the potential for dislocation core structure and its migration barrier.

The proposed strategy for transferability analysis (Figs. 6 and 7) provides a qualitative estimate of the potential performance. The outlier-based analysis can indicate if the information necessary for modelling certain defects is missing in the potential database. To improve the performance of the tested ML potential for the systems with pronounced outliers (Fig. 7a, b), their structures should be added to the potential database. At the stage

**Fig. 6 Workflow for transferability analysis of kernel ML potentials using outlier detection.** The structural data are represented in the feature space of atomic descriptor similar to that originally used to design the potential[50]. The first step of the analysis (upper panel) implies detection of defects both in the potential database and in the atomic systems to examine using MCD, one-class support vector machine (OCSVM) or any other relevant method. The second step (lower panel) is aimed at transferability analysis of the potential. The detected defect clusters from the potential database form a new training data set with the structures-inliers known by the potential. The new outlier detection model is trained on these configurations using a kernel method, e.g., OCSVM, with the kernel function identical to that of the tested ML potential. Identification of atoms-outliers within the examined defect cluster implies that these atomic environments are missing in the training database and, therefore, the tested ML potential may provide poor energetic properties for this defect.

of the potential development, the proposed defect detection protocol coupled with ML outlier detection methods (Fig. 6) can be used to optimize the content of the database, to improve the potential accuracy for modelling targeted defects and their properties.

## Discussion

This work suggests a definition of defects in crystalline solids using the distortion score of atomic environments provided by the means of distance-based ML outlier detection, notably by robust MCD. Each atom in the analysed system is described by a distortion score, which corresponds to the statistical distance of its LAE in the descriptor space from the distribution of LAEs in the reference structure. The reference structures to learn is a user choice, driven by the objectives to achieve. In this work, we have mainly employed as reference the defect-free bulk structures with some noise around perfect atomic positions.

We have numerically demonstrated that the atomic distortion score, which is based solely on geometrical information, is correlated with the local atomic energies. This finding opens up many perspectives in the field of computational materials science, with several promising applications, ranging from the qualitative substitution of the concept of energy per atom to the selection of the relevant structural information in materials design.

The present study proposes significant improvement of methods relevant for different fields of materials science and demonstrates the possibilities to overcome some blocking points in (i) structural analysis; (ii) design of new ML potentials and transferability analysis of existing ones; and (iii) advanced numerical modelling and characterization of energy landscapes.

The defect detection strategy using the distortion score is universal, i.e., in contrast to conventional geometry-based methods, it performs well for defects of a different origin. The same ML technique can be applied for the detection and analysis

of dislocations, interstitial atoms, vacancies and other defects. The proposed definition of defects through the distortion score can be used to analyse the output of various numerical methods such as massive atomistic MD (see Supplementary Note 2), Monte Carlo, metadynamics, hyperdynamics and free energy simulations. Moreover, the distortion score can be used to control the degree of precision for the relevant information to be extracted and stored. This metric can serve as a fingerprint for filtering databases with atomic structures to select and/or classify defects.

The proposed definition of defects serves to reinforce not only the performance of traditional approaches, but also of modern ML methods in materials science. Here we have demonstrate how the new concept of defects can be effectively applied for the analysis of kernel ML potentials and their databases. This approach allows optimizing the database content in order to improve the potential accuracy for modelling targeted defects and their properties. This type of potentials is able to approach DFT accuracy and can cope with large systems where the computational cost beyond the scope of ab initio methods. Improvement of these potentials can enable accurate calculations of such important physical properties, as formation and migration energy of large defects, e.g., straight dislocations and kink pairs, loops, large 3D clusters, etc. In the perspective, similar approaches can be applied to large biological/chemical molecules.

The distortion score can be applied for characterization of energy landscapes. Here, using the stratified definition of defects via distortion score, we identified the atoms with the most important contribution to the mean force of the system. Using this strategy allowed to accurately reconstruct the migration barriers from the mean force calculations of complex interstitial clusters and screw dislocations. Such an approach is of particular interest for defect localization in the simulations such as QM/MM, where the definition of total energy is ambiguous. Furthermore, the link between the distortion score and local energy

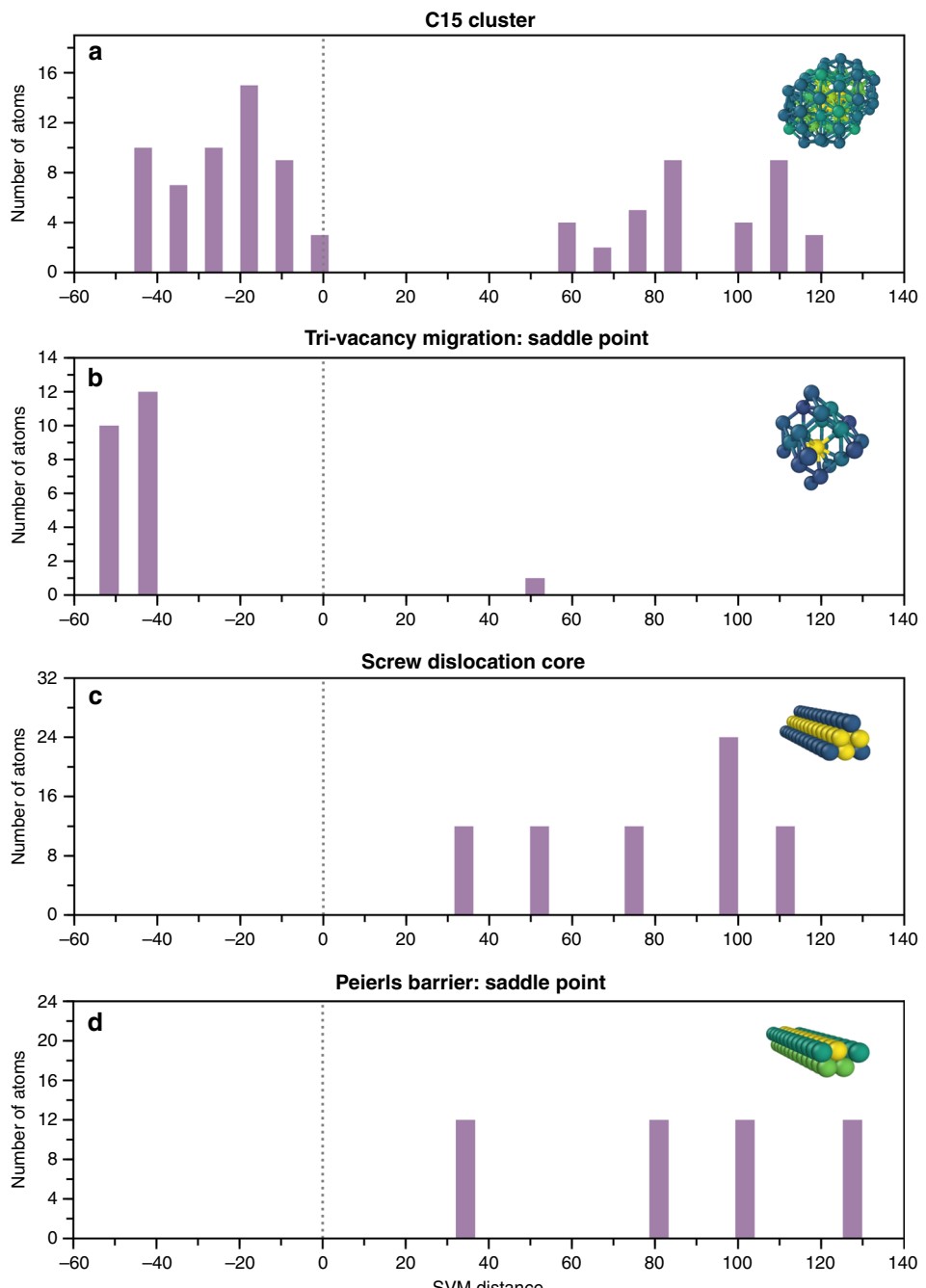

**Fig. 7 Qualitative estimation of a kernel potential performance for given defects.** Histogram of the number of atoms vs. the $d_{SVM}$ distance is plotted for the four defects not included in the database of the tested GAP potential[50] for bcc Fe, for which: **a**, **b** the potential exhibits a limited transferability; **c**, **d** the potential performs well. The inset structures from subplots (**a**–**d**) illustrate the cores of defects detected using the distortion score for the $I_{2,3}^{C15}$ cluster, the saddle point configuration of tri-vacancy migration, the minimum energy and the saddle point configurations of screw dislocations, respectively. The vertical dotted line indicates the decision boundary between the outliers (negative values) and the inliers (positive values).

opens up many perspectives for advanced MD techniques. By now, the utility of popular methods for accelerated MD, such as metadynamics[7] or mean force[8,9], statistical learning approaches[11] and temperature-accelerated dynamics/hyperdynamics[61], is often hindered by the general inability to extract the relevant information about the defects or by the definition of collective variables that are needed to compute free energy landscapes. The suggested strategy for the identification of the high-energy atoms can serve to find an appropriate reaction coordinate. This promising application has a very broad interest for the materials science community and can be further developed for the

communities of chemistry or biology, e.g., it can be applied for automated simulation schemes combined with ab initio sampling strategies.

In perspective, the notion of the distortion score based on statistical distances can be extended beyond the structural properties of defects and numerical methods of materials characterization. The present concept can be useful for the organization and the classification of multivariate data provided by experimental techniques, where the atomic coordinates are provided, such as atom probe or transmission electron microscopy tomography.

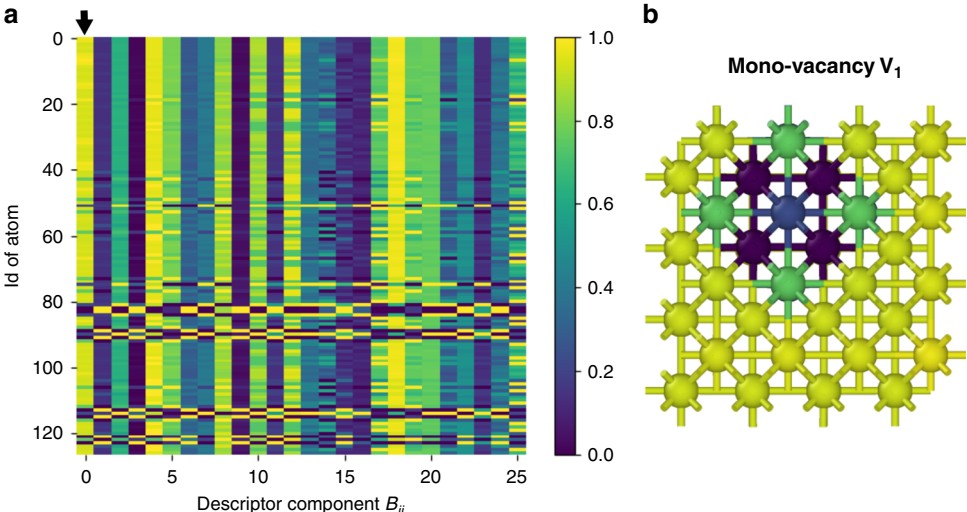

**Fig. 8 Mono-vacancy in bcc Fe represented in the descriptor space.** The 127-atom cell is represented based on the bispectrum SO(4) with $R_c = 4.0$ Å and $j_{max} = 3.5$ resulting in 26 descriptor components. **a** Heatmap of the bispectrum components. The lines deviating from the rest of the map correspond to the atomic environments impacted by the mono-vacancy. **b** Representation of the atomic array based on the first descriptor component. The atoms are coloured according to the first column, indicated with an arrow in **a**. Purple atoms correspond to the first coordination sphere of the vacancy, blue corresponds to the second and green corresponds to the third.

## Methods

**Representation of structural data and training data sets**. In this work, the training and test structural data are represented in the feature space of atomic descriptors. All atomic descriptors are calculated using the MiLaDy package[49]. Below we provide the details about atomic descriptors and the training data sets for each application presented in this study.

For the Application 1, the structural data are represented using spectral atomic descriptor, bispectrum SO(4)[36], with the angular moment $j_{max} = 3.5$ and only the diagonal bi-spectral components, which results in $D = 26$ descriptor components, as was previously described in ref. [49]. Using this representation, each atomic system with a structural defect becomes a $N \times 26$ matrix, with $N$ being the number of atoms in the simulation cell. For bcc Fe and fcc Al, we employ the cutoff distance of the descriptor function $R_c = 4.0$ Å and $R_c = 5.0$ Å, respectively, which is sufficient to take into account the nearest distorted zone around the defects. Figure 8 illustrates a 127-atom bcc Fe system with mono-vacancy represented in such a descriptor space.

The training data sets for the defect detection consist of defect-free bcc Fe and fcc Al systems. Overall, the defect detection models are trained on ca. $M = 16,200$ LAEs for each structural type. Thus, the training data sets with the bulk structures become $16,200 \times 26$. The training bulk structures contain some random noise within the Gaussian distribution with the standard deviation $\sigma = 0.08$ Å of atomic displacements, which was applied to the perfect atomic positions. Including configurations with noise into the training data set allows to prevent sensitivity of the model to atomic perturbations from their perfect positions.

For the Application 2, the reconstruction of the $I_2^{C15} \rightarrow I_2^{NP}$ SIA transition barrier in bcc Fe (Fig. 5a, b) is performed using the descriptors and training structures identical to those, applied for the Application 1. Reconstruction of the Peierls barrier (Fig. 5c, d) requires an accurate description of the long-range displacement field within the bulk structure of a material, which is not localized around the dislocation lines. Therefore, reconstruction of the barrier requires a very accurate description of any marginal perturbations within the bulk structure. To ensure a proper description of the displacement field produced by dislocations, we employ bispectrum SO(4)[36] with the angular moment $j_{max} = 4.0$ and $R_c = 5.0$ Å, and use the diagonal and non-diagonal components, i.e., $D = 55$ descriptor components per atom. In this case, we find that the structural description provided by $j_{max} = 4.0$ is sufficient to capture the subtle structural details (see comparison with $j_{max} = 4.5$ in the Supplementary Note 3). The defect-free training data set is formed by MD calculations of bcc Fe at 300 K at constant volume of 0 K using the same interatomic potential[44], as was used to compute the migration profile of dislocations. The training data set consist of $M = 25,800$ atomic environments. In the case of dislocations, employing proper MD calculations to generate the training data are preferable to application of random noise to perfect structures, as it allows to ensure an accurate description of the subtle changes in the bulk structure.

For analysis of the GAP potential transferability in the Application 3, we represent the structural data using smooth overlap of atomic positions (SOAPs) descriptor[36] with $n_{max} = 12$ and $l_{max} = 12$ for radial and angular channels, respectively, which results in dimensionality $D = 1,014$. The cutoff distance is set to

$R_c = 5.0$ Å. The same form of the SOAP descriptor was used to design the GAP potential[50].

The detection of defect clusters in the GAP database[50] is performed on the ca. 100,000 test atomic environments. After performing the outlier detection to isolate structural defects of the database, we consider ca. $M^{def} = 17,300$ atomic environments as belonging to defects. These $M^{def}$ atomic environments form the training data set (Fig. 6) for transferability analysis of the potential.

The structural data for analysis of the correlation between statistical distances and energy per atom from GAP potential in Fe (Fig. 2 and figures in the "Methods" section below) is represented with bispectrum SO(4) using $R_c = 5.0$ Å and $j_{max} = 4.5$ with all bi-spectral components. The correlations in W (in the "Methods" section below) are examined using bispectrum SO(4) with $j_{max} = 4.5$, resulting in dimensionality $D = 70$ and $R_c = 4.7$ Å, which correspond to the descriptor settings of the linear ML (LML) potential used to compute the local energies. For Fe, the training bulk structures contain ca. 103,000 atomic environments from MD calculations at 300–800 K at the constant volume of 0 K using the GAP potential. In case of W, the training was performed on 40,500 atomic environments from MD calculations at 800 K using the corresponding LML potential.

**Choosing an optimal outlier detection method**. In this work, we intend to use such an outlier detection method that not only performs well for a binary distinction between inliers and outliers but also provides a smooth decision function, which correctly reflects the detailed structure of the training and test instances. In general, density-based and clustering methods are not well adapted for the subject of the paper.

The most suitable methods should: (i) provide a smooth decision function or a similarity measure for each data point (atomic environment) with respect to the reference data cloud (e.g., defect-free structures), which can be used as a distortion score and a reliable measure of LAEs; (ii) be adapted for multivariate data sets with dimensionality from few tens (typical for the atomic descriptors used in the Applications 1 and 2) to few thousands (typical for the atomic descriptors coupled with the tested GAP potential in the Applications 3); (iii) be fast (not slower than atomistic calculations themselves) and possible to use for large systems (e.g., atomic arrays with few million atoms)—we decided to avoid methods based on non-linear kernels, as their learning process requires $M^3$ numerical operations; and (iv) be easy to implement and use for researchers from materials science community who are not necessarily experienced in ML.

Computing statistical distances is fast and more straightforward than using NNs and SVM. Moreover, there is no need to optimize hyperparameters (e.g., via grid search combined with error minimization procedures). In addition to that, it was previously demonstrated in the literature[62–64] that in some cases with relatively poorly sampled learning space, recognition of outliers can be better performed using Mahalanobis distances than with SVM and NNs. For the applications reported in our study, it is possible that the amount of available structural data for training is limited (for instance, when the data are generated from costly ab initio calculations), which can yield the situations similar to those described in refs. [62–64]. In addition to these arguments, we compare the ability of MCD and linear SVM to provide the distortion score of LAEs by measuring the correlation with the local

energy and examine their ability to provide detailed stratification of complex defects. The results are reported in the Supplementary Note 1. For both applications, MCD exhibits a better performance. For the reasons listed above, in this study we have opted to define the distortion scores based on Mahalanobis distance and robust statistical distance variants, such as robust MCD and Hotteling's distance $T^2$. These distances also were used for data mining and advanced analysis in medical and industrial applications (see the references of the review papers[26,27]).

**Minimum covariance determinant**. The strategy of outlier detection using MCD consists of computing a statistical distance from each observable to the centre of the data cloud[27,65]. An outlier is then defined as a point with a statistical distance larger than some critical cutoff. In order to describe the distance from the centre of the data and take into account the shape of the cloud, one should consider the contribution of the statistical sample covariance matrix. A classical estimator of $i_\star$ data point distance, among $M$ data points, is the Mahalanobis distance based on the sample covariance matrix $\Sigma_M \in \mathbb{R}^{D \times D}$:

$$d_{MAH}\left(\mathbf{x}_{i_\star}\right) = \sqrt{(\mathbf{x}_{i_\star} - \langle \mathbf{x} \rangle)^T \Sigma_M^{-1}(\mathbf{x}_{i_\star} - \langle \mathbf{x} \rangle)} \qquad (4)$$

The Mahalanobis distance $d_{MAH}(\mathbf{x}_{i_\star})$ describes how far is the point $\mathbf{x}_{i_\star}$ from the centre $\langle \mathbf{x} \rangle$ of the data cloud, taking into account the shape of the data distribution via $\Sigma_M$.

However, as was previously discussed in refs. [27,65], the estimators based solely on Mahalanobis distance may fail to detect mild outliers. To improve the performance of the method and annihilate the effect of outliers on the sample covariance matrix and, consequently, on the distance estimator, the so-called robust MCD estimator is used:

$$d_{RB}(\mathbf{x}_m) = \sqrt{(\mathbf{x}_m - \hat{\boldsymbol{\mu}}_0)^T \hat{\Sigma}_{M_0}^{-1}(\mathbf{x}_m - \hat{\boldsymbol{\mu}}_0)^T} \qquad (5)$$

where $\hat{\boldsymbol{\mu}}_0$ and $\hat{\Sigma}_{M_0}$ are the MCD estimates of the data cloud centre and of the MCD statistical covariance, respectively[26]. Within the MCD formalism, the whole sample covariance matrix $\Sigma_M$ is approximated by the covariance matrix $\Sigma_{M_0}$ of a data subset with $M_0 < M$ points, for which the determinant of the sample covariance matrix is minimal. The exact MCD calculation is laborious and implies computing $C_M^{M_0}$ determinants. In this work, we use FAST-MCD algorithm[37], one of the most efficient, robust and widely used version of MCD estimator[27,65]. The MCD has the ability to exclude outliers from the reduced covariance matrix, and, consequently, to increase the norm of the outliers points. MCD is an affine equivariant estimator, i.e., the data might be rotated, translated or rescaled (e.g., due to a change of the measurement units) without affecting the outlier detection diagnostics[27]. This makes MCD particularly suitable for the tasks of structural analysis. In this work, we employ robust MCD distance $d_{RB}$ (Eq. (5)) as a measure of local atomic distortion score to detect and analyse the defect structures. The outlier detection with MCD is performed on the structural data sets (see Representation of the structural data section) with contamination factor $\nu = 0.07$.

It should be noted that MCD is designed for the data with a unimodal distribution. Practically, it means that the model can be directly trained for detection of defects embedded in the structure with unimodal distribution of LAEs, e.g., in bcc anf fcc cubic metals. In order to train the model on more complex structural data with multimodal distribution of LAEs, calculations of a multidimensional distortion score can be enabled by modal decomposition of the training database. For instance, a multimodal training database $\mathcal{D}$ can be decomposed in various unimodal sub-databases $\mathcal{D}_1 \oplus \mathcal{D}_2 \oplus \ldots \oplus \mathcal{D}_n$ and a statistical distance can be computed with respect to each sub-database $\mathcal{D}_i$, providing thus an $n$-dimensional distortion score. Supplementary Note 2 provides an example of the training database decomposition and demonstrates the utility of multidimensional distortion score for the analysis of complex structural damage produced by displacement cascades.

**Statistical distances and their QM-inspired variants**. From mathematical point of view, there is a similitude between the formalism that describes the local atomic energy of materials in quantum mechanics (QM) and the statistical distances based on sample covariance matrix. As emphasized in Table 1, the observables to be evaluated are the energy of the quantum state $|i_\star\rangle$ and the statistical distance of the data point $|\mathbf{x}_{i_\star}\rangle$ in descriptor space. The local orbital basis $\{|i\rangle\}$ is equivalent to the learning database $\{|\mathbf{x}_m\rangle\}$ of the $M$ atomic environments. The eigenelement of the Hamiltonian $\{\epsilon_m, |m\rangle\}$ and $\{\lambda_m, |\mathbf{v}_m\rangle\}$ of the sample covariance matrix have similar meanings, giving the total energy (Eq. t.5) and the trace of the sample covariance matrix as the total variance (Eq. t.6). The difference here is that the occupation of each state follows a specific statistics, i.e., in QM the electrons obey Fermi-Dirac occupation $n(\epsilon)$, whereas in statistics the occupation is $n(\lambda) = 1$ for all sample points. The similar definition of global quantities, energy and variance, suggests the similar definitions of local density of states (Eqs. t.7 and t.8).

Moreover, the Eqs. t.9 and t.10 suggest that local energy and the statistical distance measure the contribution of square amplitude of probabilities of the entire spectrum of $\mathbf{H}/\Sigma_b$, which define the Hilbert space of the problem given by the Hamiltonian or sample covariance matrix, respectively, projected on measured state. The sum is weighed with the $\epsilon n(\epsilon)$ and with the inverse of the variance (the precision) in the case of electronic structure and of statistical distance, respectively. The completeness of the Hamiltonian basis gives the capacity of the model to predict new states. The similar situation concerns the statistical distance. The reliable estimation is obtained for a complete or exhaustive collection of points $\{|\mathbf{x}_m\rangle\}$ that define the sample covariance matrix.

Based on this observation, we introduce an array of statistical distances that use various weights, such as powers of eigenvalues of the sample covariance matrix, to approach the corresponding values from QM. For example, the QM of classical fermions (high temperature or $\beta \to 0$) suggests a weight similar to observable that gives the local energy and implies using $\lambda^\alpha \exp(\beta\lambda)$ instead of $1/\lambda$, where $\alpha$ and $\beta$ are constants to determine. Here we propose statistical distances with the following functional form:

$$d_{i_\star} = \left[\int d\lambda \rho_{i_\star}(\lambda)\lambda^\alpha e^{\beta\lambda}\right]^\gamma = \left[\sum_m \lambda_m^\alpha e^{\beta\lambda_m}|\langle i_\star|m\rangle|^2\right]^\gamma \qquad (6)$$

The standard MCD distance/Hotteling's $T^2$ estimator is given by the parameters $\alpha = -1, \beta = 0, \gamma = 0.5$. In case when the reference local energies are available, the parameters $\alpha$, $\beta$, $\gamma$ can be set to some optimal values. The standard choice and few sets of optimal values of these parameters for the proposed array of statistical distances are presented in the Fig. 9 for Fe and W, using two ML formalisms: GAP[50] and LML[49]. It is interesting to note that the distances inspired by the QM formalism (Fig. 9b–d, f–h) slightly outperform the standard MCD distance/Hotteling's $T^2$ estimator (Fig. 9a, e) to provide a better correlation with local energies. In this work we gave a preference to standard MCD robust distances, which do not require any information about the energy of the system.

The perspective of using a more complex function for the weight factor can be further generalized using statistical distances defined in the framework of kernel formalism. For example, with the procedure proposed in ref. [66], the authors make use of the advantages of kernel whitening and kernel PCA to compute Mahalanobis distance in the feature space by projecting the data into the subspace spanned by the most relevant eigenvectors of the covariance matrix. This extension can entirely recover the kernel formalism that underlies the GAP potential and can potentially improve the estimation of LAEs via distortion score. In conclusion, with the above considerations we found the distortion score based on various statistical distances as appropriate for measuring the distortion score of the LAEs. It worth to note that this procedure does not require any information about the energy of the system, making this conjecture particularly useful and surprising. Furthermore, when the information about the local energies is available, we propose a procedure to improve this conjecture (Eq. 6, Fig. 9).

**One-class support vector machine**. One-class support vector machine (OCSVM)[29] is a subclass of widely used support vector machine methods[67]. This approach separates inliers from outliers by finding a maximal margin hyperplane between them[29,67,68]. The vectors that determine the optimal separating hyperplane are called support vectors. OCSVM is similar to binary SVM classification, where the regular training data with the bulk structure (inliers) belongs to the first class, and the defects (outliers) belong to the second class. The proportion of outliers that contaminate the database, $v$, is an input parameter. The hyperplane between the two classes is the decision boundary, which can be defined both for linearly separable data and more complex non-linear cases.

For linearly separable data, the hyperplane can be described by the classification rule:

$$f(\mathbf{x}_m) = \langle \mathbf{w}, \mathbf{x}_m \rangle + b, \qquad (7)$$

where $\mathbf{w}$ is the normal vector and $b$ is a bias term. Both parameters $\mathbf{w}$ and $b$ are learned from the positive class (bulk) database. For each point $\mathbf{x}_m$, the value $f(\mathbf{x}_m)$ is determined by evaluating on which side of the hyperplane it falls on (in feature space). The function is positive for the inlier data points (bulk structures) and negative for the outliers (structural defects). The distance $d_{SVM}$ from the origin to a point $\mathbf{x}$ along the direction $\mathbf{w}$ is given by:

$$d_{SVM} = \mathbf{w}^T\mathbf{x}/(\mathbf{w}^T\mathbf{w})^{1/2}. \qquad (8)$$

Similarly to the MCD robust distance $d_{RB}$, the distance $d_{SVM}$ can be used as the metric of the distortion score for each atom.

In order to perform non-linear classification and obtain more complex decision boundaries, the kernel trick can be applied, as was originally proposed by Vapnik[67]. In this case, the data are implicitly mapped into a high-dimensional space through a non-linear function $\Phi(\mathbf{x})$. The distance between the data points in the new non-linear space is then measured using a non linear kernel $K(\mathbf{x}_m, \mathbf{x}_{m'}) = \Phi(\mathbf{x}_m) \cdot \Phi(\mathbf{x}_{m'})$. In this higher dimensional space the data points become linearly separable and the above linear formalism (Eq. (7)) can be applied. Most common non-linear kernels have a Gaussian (radial-basis function) or a polynomial form. For the Gaussian kernel:

$$K(\mathbf{x}_m, \mathbf{x}_{m'}) = \exp(-\gamma(\| \mathbf{x}_m - \mathbf{x}_{m'} \|)^2), \qquad (9)$$

**Table 1 Comparison of the quantum mechanics (QM) and machine learning (ML) formalism.**

| QM | ML |
|---|---|
| Consider an archetypal case, which does not reduce the generality, a solid with one orbital $|i\rangle$ per atomic site $i$. In tight-binding formalism using the hopping integrals $t_{ij}$, the probability of transition from the orbital $|i\rangle$ to orbital $|j\rangle$, the Hamiltonian reads: $\mathbf{H} = \sum_{i,j} t_{ij} |i\rangle \langle j|$. The energy levels of the system are the eigenvalues of Schrödinger equation: $\mathbf{H}|m\rangle = \epsilon_m |m\rangle$. We are interested in the estimation of the local energy $\epsilon_{i_\star}$ associated with the atom $i_\star$. | Consider that we have learned the sample covariance matrix $\mathbf{\Sigma}_b$ of $M$ data points, $x_m \in \mathbb{R}^D$. The data are centred to mean zero. The $m^{\text{th}}$ element of the descriptor space can be written in an initial basis as $|\mathbf{x}_m\rangle = \sum x_m^i |i\rangle$. The eigenelement of $\mathbf{\Sigma}_b$ is $\{\lambda_m, |\mathbf{v}_m\rangle\}$. We are interested in the statistical distance $d_{i_\star}$ of the data point $|x_{i_\star}\rangle$. |

$$\mathbf{H} = \sum_{i,j} t_{ij} |i\rangle \langle j| \tag{t.1}$$

$$\mathbf{\Sigma}_b = \sum_{i,j} \left( \frac{1}{M-1} \sum_m^M x_m^i x_m^j \right) |i\rangle \langle j| \tag{t.2}$$

$$\mathbf{H} = \sum_m \epsilon_m |m\rangle \langle m| \tag{t.3}$$

$$\mathbf{\Sigma}_b = \sum_m \lambda_m |\mathbf{v}_m\rangle \langle \mathbf{v}_m| \tag{t.4}$$

$$E = \sum_m \int d\epsilon\, n(\epsilon)\epsilon \delta(\epsilon - \epsilon_m) \tag{t.5}$$

$$Tr(\mathbf{\Sigma}_b) = \sum_m \int d\lambda\, \lambda \delta(\lambda - \lambda_m) \tag{t.6}$$

$$\rho_{i_\star}(\epsilon) = \sum_m |\langle \mathbf{i}_\star |m\rangle|^2 \delta(\epsilon - \epsilon_m) \tag{t.7}$$

$$\rho_{i_\star}(\lambda) = \sum_m |\langle \mathbf{x}_{i_\star} |\mathbf{v}_m\rangle|^2 \delta(\lambda - \lambda_m) \tag{t.8}$$

$$\epsilon_{i_\star} = \int d\epsilon\, \rho_{i_\star}(\epsilon)\epsilon n(\epsilon) \tag{t.9}$$

$$d_{i_\star}^2 = \int d\lambda\, \rho_{i_\star}(\lambda)\frac{1}{\lambda} \tag{t.10}$$

$$p(\epsilon_{i_\star}) \propto \exp(-\beta \epsilon_{i_\star}) \text{ for } \beta \to 0 \tag{t.11}$$

$$p(\mathbf{x}_{i_\star}) \propto \exp(-d_{i_\star}^2/2) \tag{t.12}$$

Commonly used quantum mechanics (QM) formalism of local energies (on left) is compared with ML formalism of sample covariance matrix and statistical distances (on right). To emphasize the similarities between the two approaches, we adopt the QM bra-ket notation for statistical distance. The data points of the descriptor space are the ket vectors $|\mathbf{x}\rangle = \mathbf{x} \in \mathbb{R}^{D \times 1}$, whereas the bra vectors are the transposed vectors $\langle \mathbf{x}| = \mathbf{x}^T \in \mathbb{R}^{1 \times D}$. $\rho_{i_\star}$ and $\rho_\lambda$ are the local density of states and variance, respectively, for the state $|i_\star\rangle$ / data point $|\mathbf{x}_{i_\star}\rangle$. $p(\epsilon_{i_\star})$ is the probability of the state $|i_\star\rangle$ in the limit of high temperature, where the Fermi-Dirac distribution becomes classical Boltzmann distribution. $p(\mathbf{x}_{i_\star})$ is the marginal likelihood of the data point $|\mathbf{x}_{i_\star}\rangle$.

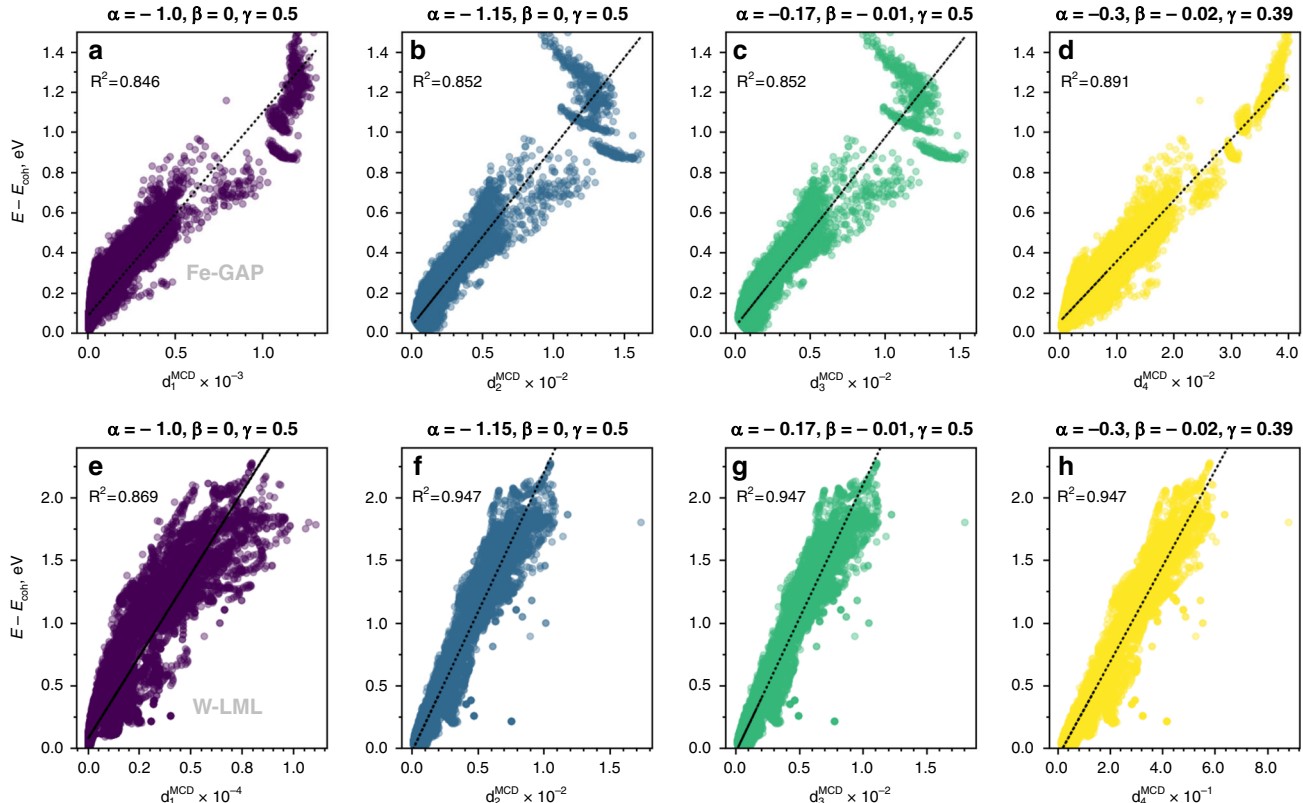

**Fig. 9 Correlation of the local energy with various statistical distances.** Correlation with the local energy from (**a–d**) GAP interatomic potential for Fe and (**e–h**) linear ML (LML) interatomic potential for W. The subplots (**a, e**) illustrate the standard MCD/Hotteling's $T^2$ estimator; (**b–d**) and (**f–h**) correspond to the variations of statistical distances inspired by QM.

where $\| \mathbf{x}_m - \mathbf{x}_{m'} \|$ is the Euclidean distance between the two data points in the descriptor space; $\gamma > 0$ is a free parameter that determines the width of the Gaussian Kernel. For Polynominal kernel:

$$K(\mathbf{x}_m, \mathbf{x}_{m'}) = (\gamma(\mathbf{x}_m \cdot \mathbf{x}_{m'}) + c)^p, \tag{10}$$

where $p$ stands for the $p$-degree of the polynomial, $c \geq 0$ is a parameter that controls the influence of higher-order vs. lower-order terms in the polynomial and $\gamma$ is a hyper parameter.

In this work, the structural analysis of defects is performed using OCSVM with Gaussian kernel with $\gamma = 0.03$. For the transferability analysis of the GAP potential, we employ a polynomial kernel identical to that, which was originally used for the design of the ML potential[50], i.e., with $p = 4$ and $c = 0$ (homogeneous kernel). With this choice of the kernel parameters, $\gamma$ is a scaling factor that impacts the magnitude of the distances between configurations. For transferability analysis of the potential, contamination factor $\nu$ (the upper bound on the fraction of training errors and a lower bound of the fraction of support vectors) is set to $10^{-3}$, to obtain a tight decision boundary.

**Reporting summary**. Further information on research design is available in the Nature Research Reporting Summary linked to this article.

## Data availability

The training databases for Fe and Al as well as the analysed configurations are available in public GitHub repository at https://github.com/mcmarinica/DefectsDetection.

## Code availability

The descriptors for various structures were computed using `MiLaDy` package and the structural analysis was performed using `Unseen` package. The relevant codes to reproduce the results presented in this paper are available upon request from the corresponding authors.

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

## Acknowledgements
This work was financially supported by the Cross-Disciplinary Program on Numerical Simulation of CEA, the French Alternative Energies and Atomic Energy Commission. A. M.G., C.L., J.D. and M.C.M. acknowledge the support from GENCI - (CINES/CCRT) computer centre under Grant number A0070906973. This work has been carried out within the framework of the EUROfusion Consortium and has received funding from the Euratom research and training programme 2014–2018 and 2019–2020 under grant agreement number 633053. The views and opinions expressed herein do not necessarily reflect those of the European Commission. This work also received funding from the Euratom research and training programme 2019–2020 under grant agreement number 755039. A.M.G. acknowledges Marie Landeiro dos Reis for providing simulation cells with dislocations in fcc Al.

## Author contributions
A.M.G. and M.C.M. designed the study. M.C.M. and J.B.M. supervised the study. A.M.G. performed the structural analysis. C.L., C.D. and J.D. performed atomistic calculations. All authors participated in discussion and interpretation of the results. A.M.G. and M.C.M. wrote the manuscript.

## Competing interests
The authors declare no competing interests.
