## [Peer Review File · Nature Communications]

Reviewers' comments:

Reviewer #1 (Remarks to the Author):

This paper presents results of analysis of atomistic models of defect structures in crystalline solids. The analysis employs an innovative combination of two distinct technologies: rotationally invariant descriptors of local atomic environments and data analytic outlier detection (distortion score). By combining these two techniques, the authors demonstrate improved discrimination of different kinds of defects relative to conventional methods such as local atomic energy and common neighbor analysis. In addition, the authors identify a strong correlation between local atomic energy and the data analytic distortion score. With multiple examples, the authors demonstrate that the distortion score can be used to effectively identify the atoms involved in a defect structure, without the need to estimate local atomic energy. The usefulness of the approach indicates that it will achieve widespread adoption in the rapidly growing fields of computational materials design, machine learning interatomic potential development, and multiscale materials modeling.

Overall the quality of the work and the writing is high and I recommend publication without revisions.

Reviewer #2 (Remarks to the Author):

The paper introduces Minimum Covariance Determinant (MCD) to characterize and classify local atomic environments. The methodology is interesting but I think, it is not rather unique in the same that methods which are rotational invariant have been around since the first implementation of genetic algorithms in structural search. I fully understand that the MCD is maybe more well supported but I really do not see the great deal that the authors present in the paper. I understand that the use of well defined local measurements can improve some existing methods, even the performance of the GAP potentials and some other problems but I do not see the contribution as relevant to be published from Nature Communications. I think this paper should go to a more focus journal, where this effort can be more appreciate it.

Additionally to my previous comment, I have also some disagreement with respect to the discussion the authors made with respect to the local energy. In DFT, that is not possible (EVEN if the Kohn-Sham orbitals are projected into atomic orbitals), though DFT is a local theory, it is over the density. Making the parametrization of the bands with Tight-Binding methods is possible to do this distinction but, to my knowledge, it has not been proven that this reproduces the same significance than those obtained by using local spatial operators in DFT. I agree that it is very tempting to consider the energy partition obtained from Machine Learning methods as a good approximation to the energetics of the system, but I do not consider that as a general truth. At the end of the end, ML will not do more than what it is taught, basically it will not do better than DFT and in DFT you have intrinsic problems, you will reproduce them from any ML method.

Reviewer #3 (Remarks to the Author):

The paper's concerned with applying outlier detection methods to aid in the detection, description, and modeling of defects in crystalline solids. The authors demonstrate how an outlier detection method based on distance score to a point clouds center, MCD, can be used to find certain defects, report how it correlates to local atomic energy, how to calculate the mean force, and how to assess whether Gaussian-process-based potentials accurately capture defect characteristics.

I know next to nothing about the physics or crystals, or material science, so I'll only comment on the ML part.

As such, while I found the work described in the manuscript rather convincing, there is a bit of a tension between the very general claims, such as "The proposed definition of defects opens many perspectives in materials design and high-throughput techniques for large data set generation." and the very concrete focus on MCD.

Concretely, apart from pointing out that MCD can deal with data that can be rotated, translated or rescaled, there isn't any justification for the choice of the method itself to be found. The field of outlier detection is very rich, something that the authors touch on when they discuss related work, so a discussion of why MCD, or even better an experimental evaluation of different methods would be good.

I also mention this because the authors state early on: "It is worth mentioning that MCD is designed for unimodal distribution. Consequently, a careful selection of the training data should be performed to avoid multimodal distributions."

Maybe there are other methods that don't have that drawback? And will this type of training data selection interfere with the usefulness of the method? At the end of the manuscript, this returns, this time in the form: "Practically, it means that the model can be trained for detection of defects within one structural type, e.g., hcp domain within fcc structure will be identified as a defect." - does this mean that MCD will be unable to identify different defects if they occur in the data together?

My second issue is that there are a number of parameters involved, the values of which are never justified. There's for instance the dRB threshold, which decides which atoms are identified as outliers. From what I've seen in the text, this threshold takes the values 3.9, 12, 17, 2.9. Deciding the threshold above which an outlier score is interpreted as actually indicating an outlier is a far from trivial problem in outlier detection and a number of strategies have been proposed (starting from relatively straight-forward extreme values approaches to more complex methods, e.g. iterative ones). Giving guidance to the practitioner as to how to set this threshold will influence how useful/successful the use of outlier detection is.

This is also the case, although less problematic, for the parametrization of the SVM. The authors set γ for the kernel to 0.03, without explaining why. The value of the complexity parameter C is not mentioned. For the transferability analysis, $p=4$ and $C=0$ seem to be justified by the values in the literature but "kernel similar to that, which was originally used for the design of the ML potential" sounds as if it's precisely not the same.

Language remarks - there's a lot of small stuff that indicates that the paper should be proof-read again:

- page 3: "the materials properties" is missing a possessive apostrophe somewhere
- page 3: "Identification and characterization of defects provides" should be "provide" since it's plural
- page 3: " and a by rapidly growing number of fast exploring, biased in energy" - apart from the fact that I suspect it should be "based in energy", this partial phrase doesn't work grammatically

I'm not going to go through everything I highlighted in the text, just, I didn't understand "The subtle elastic interaction of dislocations rises relaxation patterns that are captured by the distortion score." at all.

All in all, while I found the paper interesting and convincing, the authors need to

- 1) decide whether they want to argue in favor of outlier detection or push MCD as the method of choice. In either case, alternatives should be discussed and/or evaluated, and the choice of MCD motivated. In the latter case, some of the too general claims, as well as the title, need to be revised.
- 2) explain how one would set parameters, particularly the cutoff value of the outlier score, and how to select data to arrive at a reliable inlier model.
- 3) improve the language.

Response to reviewers

Ref. No.: NCOMMS-20-05687

Title: Reinforcing materials modelling by encoding the local structures of defects
in crystalline solids into distortion scores

Authors: Alexandra M. Goryaeva et al.

May 13, 2020

We thank the referees for the careful reading of our manuscript and for the recognition of our work, convening physics with statistical ML concepts, as novel and interesting. We are glad that the paper was reviewed by specialists both from the communities of physics and ML. The diverse comments of the referees helped us to enhance the multidisciplinary character and the impact of the paper, making it more suitable for the broad audience of Nature Communications.

In order to improve the quality of the paper and to answer the constructive criticism of the reviewers, a careful revision of the manuscript was performed. In particular, we have demonstrated the advantages of using statistical distance based outlier detection compared to other commonly used methods, like SVM, for the applications presented in this work. Moreover, in order to give a broader coverage of the subject for the to wide readership of Nature Communications, we have updated the Methods section and provided complementary analysis and examples in the Supplementary Materials. The applied changes improve the visibility of technical points that had been raised by the referees. In addition to the changes mentioned above, we have emphasised the synergy between materials science and statistical machine learning through the proposed concept of distortion scores. In particular, we have spotlighted the links between some concepts from the machine learning community with such concepts from materials science, as defects or local energy. Moreover, following this crossover of the concepts, a set of physics-inspired statistical distances is proposed .

The changes in the manuscript file are highlighted in green. Below we provide a point by point answer to each reviewer. We believe that the performed revisions have significantly improved the manuscript, and we hope that the applied changes, as well as the important extension of the Methods section and the Supplementary Materials will lead to acceptance of the paper.

Reviewer #1

This paper presents results of analysis of atomistic models of defect structures in crystalline solids. The analysis employs an innovative combination of two distinct technologies: rotationally invariant descriptors of local atomic environments and data analytic outlier detection (distortion score). By combining these two techniques, the authors demonstrate improved discrimination of different kinds of defects relative to conventional methods such as local atomic energy and common neighbor analysis. In addition, the authors identify a strong correlation between local atomic energy and the data analytic distortion score. With multiple examples, the authors demonstrates that the distortion score can be used to effectively identify the atoms involved in a defect structure, without the need to estimate local atomic energy. The

usefulness of the approach indicates that it will achieve widespread adoption in the rapidly growing fields of computational materials design, machine learning interatomic potential development, and multiscale materials modelling.

Overall the quality of the work and the writing is high and I recommend publication without revisions.

We are glad that we have convinced the referee concerning the broad importance and impact of our study, and we hope that our response to Reviewer #2 and Reviewer #3 will result in acceptance of the manuscript.

Reviewer #2

R2-1. The paper introduces Minimum Covariance Determinant (MCD) to characterise and classify local atomic environments. The methodology is interesting but I think, it is not rather unique in the same that methods which are rotational invariant have been around since the first implementation of genetic algorithms in structural search. I fully understand that the MCD is maybe more well supported but I really do not see the great deal that the authors present in the paper. I understand that the use of well defined local measurements can improve some existing methods, even the performance of the GAP potentials and some other problems but I do not see the contribution as relevant to be published from Nature Communications. I think this paper should go to a more focus journal, where this effort can be more appreciate it.

We thank the referee for finding our methodology interesting. We respect her/his opinion about the relevance of our work for a more focused journal, however, we believe that our paper has a broad interest for the community of physics and materials science and meets the criteria of Nature Communications for high quality research. According to the Aims and Scope of the journal, Nature Communications is a *multidisciplinary journal dedicated to publishing high-quality research in all areas of the biological, health, physics, chemical and Earth sciences. Papers published by the journal aim to represent important advances of significance to specialists within each field.* Our study has a broad interest for the community of physics as it proposes a significant improvement of the methods relevant for multiple fields of the community and demonstrates the possibilities to overcome some blocking points in (i) structural analysis; (ii) advanced materials modelling, and (iii) design of new ML potentials and transferability analysis of existing ones. Therefore, this work will have immediate and substantial implications for the research involving crystalline materials. Publishing in a more focused journal would imply publishing three articles focused on each of the proposed applications, which does not allow to demonstrate universality of the proposed approach and its broad applicability for the community of physics.

In addition to the broad applicability for the community of physics, this work represents a synergy between materials science and statistical machine learning. We revise the concept of defects in materials science community using the means of machine learning. The present work bridges the communities of physics and machine learning and establishes the links between the concepts from these two fields (Table 1 in this document) in such a way that have never been considered as mutually related. We have used robust MCD distances, also referred hereafter as statistical distances, that until now have been never applied for the needs of materials science, to provide the distortion score as a local measure of local atomic environments. This strategy to provide a measure of local atomic environments is not unique and probably will be the topic of further studies. The main advantage of using MCD, among the vast choices offered by ML community, is that it provides a distortion score that fulfils many criteria that are essential for the materials science community (see the answer R3-2 in the present document). Being *intrinsically correlated with local atomic energies* (Fig. 1 in the manuscript; Fig. 3, Table 1 in this document an in the revised manuscript), the notion of distortion score is a genuine advancement in the modern material science, which opens up many perspectives for various applications. Here, due to the limitations of the

article length, we were restricted to three examples: structural analysis, multiscale computations and development of interatomic potentials. We are deeply convinced that our paper represents a significant interest for the community of physics and that many perspective applications of the proposed method will be the topic of further studies.

We regret the lack of clarity in the originally submitted manuscript that did not express clear enough that the distortion score represents an advanced concept, which is significantly different from the conventional “rotational invariant methods”. After a careful proof-reading of the text, we have added some explicit explanations of the advantages of the proposed method and of its correlation with the local energy.

In order to highlight the link between computational physics and some concepts from the machine learning community and to improve the multidisciplinary impact of the paper we have made the following changes:

- In Methods, we included a new section, which explains the choice of relevant machine learning methods to provide distortion score of local atomic environments. The section “*Choosing optimal outlier detection methods to provide a pertinent distortion score*” outlines the criteria that should be fulfilled by outlier detection methods in order to be valuable for the materials science community and to provide an appropriate distortion score. Complementary to this Methods section, we included a new Section I in Supplementary materials, which provides a technical comparison between the statistical distances from MCD and other widely used ML method, Support Vector Machine (SVM).
- In addition to that, we included Methods section “*Statistical distances and their Quantum Mechanics (QM)-inspired variants to set out the distortion scores*” foregrounds the relationship between quantum mechanics and the statistical distances formalism. Moreover, in this section we propose a set of QM-inspired of statistical distances.
- In order highlight a universal character of the proposed defect detection (Application 1), we have added in Supplementary Materials a new Section II with analysis of complex structural damage produced by a highly energetic displacements cascade with more than 5 millions atoms. The aim of the section “*Multidimensional distortion score for analysis of displacement cascades and detection of targeted defect types*” is to demonstrate the performance of the proposed method for the applications where standard well-established methods of defect identification fail, e.g., to detect complex 3D structures of interstitial clusters. For this analysis, we have used the concept of *multidimensional* distortion score. With the suggested procedure we are able to identify 3D defect clusters that can not be detected with ordinary geometry-based techniques from the materials science community.

We hope that with the arguments brought forward above together with the corrections of the manuscript, we could convince the referee of the relevance of our paper for Nature Communications. In case the referee has further remarks, we would willingly engage in a more specific discussion regarding this topic.

R2-2. Additionally to my previous comment, I have also some disagreement with respect to the discussion the authors made with respect to the local energy. In DFT, that is not possible (EVEN if the Kohn-Sham orbitals are projected into atomic orbitals), though DFT is a local theory, it is over the density. Making the parametrization of the bands with Tight-Binding methods is possible to do this distinction but, to my knowledge, it has not been proven that this reproduces the same significance than those obtained by using local spatial operators in DFT.

We thank the referee for this remark. We realise that this point was possibly presented in a confusing way in Introduction of the submitted manuscript. However, we aimed claiming there the very same

point that the referee mentions concerning the local energy in PW methods. In the submitted version of the paper, we mentioned that “*energy and stress per atom cannot be directly extracted from the PW methods*”, meaning that local energy cannot be extracted without any post-treatment. Further in the text (page 10, 1st paragraph of Application 2) “*For instance, in the case of widely used plane-wave (PW) electronic structure calculations, the definition of energy per atom is ambiguous and requires to project delocalised electron density on local atomic orbitals.*”

Indeed, as the referee indicates, there are several non-trivial ways to perform post-treatment of the densities from DFT to extract the local energies, e.g., using tight binding calculations [1, 2], Green function [1, 3-5], projection of atomic orbitals in the sense of Mulliken projections [6], etc. How far these local energies will be from the “real” values and what would be their significance are the questions beyond the scope of this study.

Our paper (Application 2) mainly focuses on the calculations where *ab initio* methods are involved in such a way that even the concept of total energy can not be properly defined. The definition of energy profile is critical in many statistical learning approaches, including the considered quantum mechanics/molecular mechanics methods QM/MM, which is currently at the forefront of computational materials science [7-9]. In this method, the system commonly consists of the two parts: the core, which is described using *ab initio*; and the outer part, which follows classical mechanics or surrogate tight binding Hamiltonian (the main contribution that has fast force evaluation). The interaction of these parts and description of the whole system are given solely by the forces, which are well defined local quantities in this case. However, the total energy of the system cannot be defined in this type of calculations. Moreover, the wavefunction of the core part is highly perturbed by the buffer region between the two parts of the system, which makes the attempts to define the local energy useless. As a consequence, these methods *cannot have access neither to local nor to total energies*. In our work we suggest using distortion scores of atomic environments in order to have access to local quantities that can indicate the atoms with most important contribution based solely on geometric information. The atoms from the core and the outer part of the system are treated on the same footing. Using the distortion score as local information we are able to integrate the mean force along the complex reaction coordinate and find the migration/transformation energy barrier for systems where the energy cannot be directly defined.

After the reviewer’s remark, we realise that the way we have phrased the energy-related motivation for using the of the distortion score in QM/MM calculations (Application 2) was probably very short and confusing. In order to improve the clarity and to remove this ambiguity, we have applied some corrections of the text:

- In the Introduction we have added the following text:

For instance, energy and stress per atom cannot be directly extracted from the widely used *ab initio* plane-wave (PW) methods. In this case, a post-treatment, such as projection on local orbitals or Mulliken analysis, is needed. In some multiscale simulations, e.g., in QM/MM, even the concept of total energy is not well defined.

- In the Application 2 we have added the following paragraph:

The definition of energy profile is critical in many statistical learning approaches, including the considered quantum mechanics/molecular mechanics methods QM/MM, which is currently at the forefront of computational materials science [7-9]. In this method, the system commonly consists of the two parts: the core, which is described using *ab initio*; and the outer part, which follows classical mechanics or surrogate tight binding Hamiltonian (the main contribution that has fast force evaluation). The interaction of these parts and description of the whole system are given solely by the forces, which are well defined local quantities. However, the total energy of the system cannot be defined in this type of calculations. Moreover the wavefunction of the core part is highly perturbed by the buffer region between the two parts of the system, which makes the attempts to

define the local energy useless. As a consequence, these methods cannot have access neither to local nor to total energies.

- In order to better emphasise the utility of the distortion score in the context of Application 2, we have added the following paragraph before the equation 3 :

This simple approach is sufficient for the calculations of particular class of compact defects, like interstitial clusters, but it does not provide a universal solution, e.g., it is not applicable for the defect structures that can not be well localised, like dislocations. Here, we suggest using the distortion score to define the confidence region based solely on geometric information of LAEs. The atoms from the core and the outer part of the system are treated on the same footing. Using the distortion score as local information we are able to indicate the atoms that are more likely to contribute to the mean force of the system. Finally, we integrate the mean force along the complex reaction coordinate and find the true migration/transformation energy barrier for systems where the energy cannot be directly defined.

R2-3. I agree that it is very tempting to consider the energy partition obtained from Machine Learning methods as a good approximation to the energetics of the system, but I do not consider that as a general truth.

This paper proposes the distortion score as a measure of local structure. The distortion score can be one-dimensional but not limited to this definition. For one-dimensional distortion scores, we have numerically demonstrated a conjecture that the distances provided by MCD exhibit a correlation with local energies (Figures 2, 9 and 10 in the manuscript). In other cases (Application 2), this score serves as a collective variable. In the multi-dimensional case, the score can be used to localize the most relevant structures in complex energetic landscape (Section II of Supplementary Materials "*Multidimensional distortion score for analysis of displacement cascades and detection of targeted defect types*").

Being correlated with local energies, one dimensional score can serve as a *qualitative* surrogate of local energies, but we have never used it as a direct quantitative replacement of energy, neither of local nor of total. Distortion scores only indicate high-energy atoms, but we never claimed that this values directly correspond to the energies. In order to reconstruct the energy barriers (Application 2), we use the forces. Distortion scores in this case act as a collective variable and only indicate the atoms with the most important contribution, which should be taken into the region for mean force integration. We do not suggest to make a sum of distortion scores in order to reproduce the energy barrier. In this sense, we have never claimed that the suggested measure of local structure can quantitatively be described as something "partitioning the energy". In other words, it is a good qualitative indicator to localize important areas in the structures.

As we have stated in **R2-2**, we agree that 'true' local energy can not be computed, but the local measure computed by ML fulfills the requirements in order to serve as a criteria for evaluation of other physical properties. The suggested approach represents an excellent example of the synergy between the communities, which can reinforce materials modelling and enable access to many physical quantities and properties, such as accurate total energy and molecular dynamics trajectories [10–12], free energy sampling with *ab initio* accuracy [13] or even investigation of continuum domains of materials science [14]. All these properties are challenging or even impossible to obtain using solely the traditional methods of the community of physics.

In order to improve the clarity regarding applicability of the distortion scores, we have applied some corrections of the text:

- We have changed the title of the paper, aiming thus to express the focus of the study on the

distortion score, the major concept of the present paper; (ii) modified the abstract and reorganised the Discussion section in order to more clearly indicate purposes and applicability of the method and its perspective developments in the future.

- Added two paragraphs in the text concerning Application 2, as it is mentioned in the answer R2-2.

R2-4. At the end of the end, ML will not do more than what it is taught, basically it will not do better than DFT and in DFT you have intrinsic problems, you will reproduce them from any ML method.

We agree with the referee's claim that any form of ML methods cannot fully replace traditional approaches in physics and/or materials science. The phase space in physics / materials science has a well defined structure with states that follow the Schrödinger equation of the system's Hamiltonian and for which the dynamics is driven by the Liouville's flow. This space is too vast and complex to be properly described only by the inherent statistical correlations within the data points. To our knowledge, at this moment, none of the statistical methods alone, generically called ML and its subclass deep learning, can provide a valuable alternative to the laws of physics. In order to provide reliable results in the field of physics, ML and deep learning, should learn and be trained on the coherent data provided by well established methods from the community of physics.

Nevertheless, statistical methods trained on the physical data can be of great help when the traditional methods are limited or/and their direct application is hindered by such factors as high computational cost. In the manuscript we have exemplified three cases where ML effectively helps reinforcing the traditional approaches. The first two applications extend the applicability of methods for the defect modelling characterisation, including the quantum-classical methods like QM/MM. The third application helps profiling and improving transferability of the kernel based ML interatomic potentials. This type of potentials are able to approach DFT accuracy and can cope with large systems where the computational cost beyond the scope of *ab initio* methods. While ML approaches scale with $O(N)$ for a system with N atoms, DFT method require at least $O(N^3)$.

The synergy of ML approaches with traditional methods opens many avenues in material science giving access to the crucial physical properties, such as formation and migration energy of large defects of different origin, e.g., straight dislocations and kink pairs, loops, large 3D clusters, etc. In the perspective, similar approaches can be applied to large biological / chemical molecules.

To conclude, there is no question of replacing DFT with ML or considering results from ML superior to DFT. Statistical methods help us to overcome the limitations of traditional methods from the community of physics and to fill the gaps between the length/time scales in order to enable further progress and developments in the community.

Reviewer #3

The paper's concerned with applying outlier detection methods to aid in the detection, description, and modelling of defects in crystalline solids. The authors demonstrate how an outlier detection method based on distance score to a point clouds centre, MCD, can be used to find certain defects, report how it correlates to local atomic energy, how to calculate the mean force, and how to assess whether Gaussian-process-based potentials accurately capture defect characteristics.

I know next to nothing about the physics or crystals, or material science, so I'll only comment on the ML part.

We are glad that the manuscript was reviewed by a specialist in the ML field. Although the reviewer

mentions that he/she is not familiar with the physics of crystals, we find his/her comments both on ML part and on the physics-related part very spot on. The suggestions of the referee will definitely improve the manuscript. Below we provide our replies to the comments of the referee.

R3-1. As such, while I found the work described in the manuscript rather convincing, there is a bit of a tension between the very general claims, such as "The proposed definition of defects opens many perspectives in materials design and high-throughput techniques for large data set generation." and the very concrete focus on MCD.

We sincerely appreciate the fact the referee found our work convincing and acknowledge him/her for pointing us the tension between some general claims and a our focus on the concrete method. We realise that the way we had formulated some sentences and organised the Methods section in the original version of the manuscript was a bit confusing. In order to improve this point in the revised manuscript, we have reformulated and reorganised the ambiguous parts of the manuscript.

In order to better emphasise the utility of the distortion score and explain the focus on MCD and statistical distances, we have (i) modified the title and the abstract ; (ii) significantly extended the Methods section and Supplementary Materials (see the answers to **R3-2**, **R3-3** and **R3-4**);

In order to remove the ambiguity and possible misinterpretation of too generally formulated claims, we have reorganised the Discussion section and reformulated some phrases. In the original version of the article, "The proposed definition of defects opens many perspectives in materials design and high-throughput techniques for large data set generation" was formulated in the context of materials science concepts to convey the aims of (i) better characterisation of defects yields into better multiscale modelling and, consequently, into reliable materials design; (ii) usefulness of distortion score that is of great help to analyze huge amount of data produced by modern high-throughput techniques of advanced numerical simulations in materials science, such as massive molecular dynamics, metadynamcis, mean-force, etc. In the revised version of the Discussion, we clearly indicate the context in terms of applicability of the method and its perspective developments in the future.

R3-2. Concretely, apart from pointing out that MCD can deal with data that can be rotated, translated or rescaled, there isn't any justification for the choice of the method itself to be found. The field of outlier detection is very rich, something that the authors touch on when they discuss related work, so a discussion of why MCD, or even better an experimental evaluation of different methods would be good.

We thank the referee for indicating us the lack of clarity regarding the choice of MCD as a main tool. Following the referee's advise, we have done our best in order to clarify this aspect in the new version of the manuscript. Indeed, the field of outlier detection is very rich and, before working with MCD, we have tested several methods in order to find the most appropriate one for our purposes. However, this information was not well emphasised in the originally submitted version of the paper.

In this work, we intend to use such an outlier detection method that not only preforms well for a binary distinction between inliers and outliers, but also provides a smooth decision function, which correctly reflects the detailed structure of the training and test instances. Generally, density-based and clustering methods are not well adapted for the subject of the paper.

The most suitable methods should:

- (i) provide a smooth decision function or similarity measure for each data point (atomic environment) with respect to the reference data cloud (e.g., defect-free structures), which can be used as a distortion score and a reliable measure of local atomic environments;

- (ii) be adapted for multivariate data sets with dimensionality from few tens (typical for the atomic descriptors used in the Applications 1 and 2) to few thousands (typical for the atomic descriptors coupled with the tested GAP potential in the Applications 3);
- (iii) be fast (not slower than atomistic calculations themselves) and possible to use for large systems (e.g., atomic arrays with few million atoms). We decided to avoid methods based on non-linear kernels as their learning process requires M^3 numerical operations. In this study we employed the databases that contain from 1.5×10^4 to 10^5 atomic environments for training.
- (iv) be easy to implement and use for researchers from materials science community who are not necessarily experienced in ML.

For the reasons listed above, in this study we have opted to define the distortion scores based on Mahalanobis distance and robust statistical distance variants, such as robust MCD and Hotteling’s distance T^2 . These distances also were used for data mining and advanced analysis in medical and industrial applications (see the references of the review papers [15, 16]).

Computing statistical distances is fast and more straightforward than using neural networks (NNs) and support vector machine (SVM). Moreover, there is no need to optimise hyperparameters (e.g., via grid search combined with error minimization procedures). In addition to that, it was previously demonstrated in the literature [17–19], that in some cases with relatively poorly sampled learning space, recognition of outliers can be better performed using Mahalanobis distances than with SVM and NNs. For the applications reported in our study, it is possible that the amount of available structural data for training is limited (for instance, when the data is generated from costly *ab initio* calculations), which can yield the situations similar to those described in Refs [17–19].

Before making a decision in favour of outlier detection based statistical distances, we have tested the performance of one class SVM, one of the most widely used methods with a smooth decision function. Overall, we find that SVM has an excellent performance to detect structural outliers (binary classification to distinguish between bulk structure and defects). For this reason, we have used SVN as a final step of transferability analysis in the Application 3. However, in addition to the outlier detection, it is very important for our applications to obtain the distances that enable a detailed description of the structure in order to be used as a distortion score of atomic environments. We found that SVM distances are not well suited for detailed stratification of defects. This feature of SVM distances was shown in the Supplementary materials. However, we realise that the reference for this comparison of SVM with MCD was not clearly provided in the manuscript. In the corrected version of the paper, we make sure that the reference for this comparison clearly appears. Following the reviewer’s recommendation, we additionally examine the correlation of SVM distances d_{SVM} with local atomic energies and report the results in the Supplementary materials. Figure 1 shows the best correlation of d_{SVM} with the local atomic energies based on the same structural data as Figure 2 in the main text of the paper. Overall, linear SVM distances are positively correlated with the local atomic energies, however, the correlation coefficient R^2 is systematically lower than 0.7, which does not allow to use d_{SVM} as an accurate measure of local atomic environments. Using SVM with a non-linear kernel could help improving the correlation, however, it will lead to a drastic increase of numerical cost of the analysis and will imply a careful choice of the kernel form and parameterization. Moreover, the chosen kernel does remain universal for different types of defects and the parameterization should be adjusted by the user for the given application. These characteristics of the non-linear kernels do not satisfy the requirements (iii) and (iv) listed above, therefore we have a preference to avoid SVM with non-linear kernels as universal method of structural analysis.

The above reasons drove us towards statistical distances. Moreover, from mathematical point of view, there is a similitude between the formalism that describes the local atomic energy in quantum mechanics and the statistical distances based on sample covariance matrix.

Figure 1: Correlation of the local energy from GAP interatomic potential for Fe with linear SVM distances. Each point on the plot represents an individual atomic environment. The structural data is taken from the GAP potential database [20] and fully corresponds to that from Figure 2 in the main text of the manuscript.

Figure 2: Correlation of the local energy from GAP interatomic potential for Fe with robust statistical distances. The subplot (a) illustrates the standard MCD / Hotteling's T^2 estimator; (b, c, d) correspond to the variations of statistical distances inspired by QM.

As emphasised in Table 1, the observables to be evaluated are the energy of the quantum state $|i_\star\rangle$ and the statistical distance of the data-point $|\mathbf{x}_{i_\star}\rangle$ in descriptor space. The local orbital basis $\{|i\rangle\}$ is equivalent to the learning database $\{\mathbf{x}_m\}$ of the M atomic environments. The eigenelement of the Hamiltonian $\{\epsilon_m, |m\rangle\}$ and $\{\lambda_m, |\mathbf{v}_m\rangle\}$ of the sample covariance matrix have similar meanings. Total energy (Eq. 5) and the trace of the sample covariance matrix as the total variance (Eq. 6). The difference here is that the occupation of state follows a specific statistics, i.e., in QM the electrons obey Fermi-Dirac occupation $n(\epsilon)$, whilst in statistics the occupation is $n(\lambda) = 1$ for all sample points. The similar definition of global quantities, energy and variance, suggests the similar definitions of local density of states Eqs. 7 and 8.

Moreover, the Eqs. 9 and 10 suggest that local energy as well as the statistical distance, measure the contribution of square amplitude of probabilities of the entire spectrum of \mathbf{H}/Σ_b , which define the Hilbert space of the problem given by the Hamiltonian or sample covariance matrix, respectively, projected on measured state. The sum is weighed with the $\epsilon n(\epsilon)$ and with the inverse of the variance (the precision) in the case of electronic structure and of statistical distance, respectively. The completeness of the Hamiltonian basis gives the capacity of the model to predict new states. The similar situation concerns the statistical distance. The reliable estimation is obtained for a *complete* or exhaustive collection of

Figure 3: Correlation of the local energy from linear ML (LML) interatomic potential for W with the robust statistical distances. The subplot (a) illustrates the standard MCD / Hotteling's T^2 estimator; (b, c, d) correspond to the variations of statistical distances inspired by QM.

QM		ML	
Consider an archetypal case, that not reduces the generality, a solid with one orbital $i\rangle$ associated on the i^{th} atom. In tight binding formalism using the hopping integrals t_{ij}, the probability of transition from the orbital $i\rangle$ to orbital $j\rangle$, the Hamiltonian reads: $\mathbf{H} = \sum_{i,j} t_{ij} i\rangle \langle j$. The energy levels of the system are the eigenvalues of the Schrodinger equation: $\mathbf{H} m\rangle = \epsilon_m m\rangle$. We are interested in the estimation of the local energy ϵ_{i_\star} associated with some quantum state $i_\star\rangle$.		Consider that we have learned the sample covariance matrix Σ_b of M data points, $x_m \in R^D$. The data is centred to mean zero. The m^{th} element of the descriptor space can be written in an initial basis as $\mathbf{x}_m\rangle = \sum x_m^i i\rangle$. The eigenelement of Σ_b is $\lambda_m, v_m\rangle$. We are interested in the statistical distance d_{i_\star} of the data point $\mathbf{x}_{i_\star}\rangle$.	
$\sum_{i,j} t_{ij} i\rangle \langle j $ (1)	\mathbf{H}	Σ_b	$\sum_{i,j} (\frac{1}{M-1} \sum_m^M x_m^i x_m^j) i\rangle \langle j $ (2)
$\mathbf{H} = \sum_m \epsilon_m m\rangle \langle m $ (3)	\mathbf{H}	Σ_b	$\sum_m \lambda_m v_m\rangle \langle v_m $ (4)
$\sum_m \int d\epsilon n(\epsilon) \delta(\epsilon - \epsilon_m)$ (5)	E	$Tr(\Sigma_b)$	$\sum_m \int d\lambda \lambda \delta(\lambda - \lambda_m)$ (6)
$\sum_m \langle \mathbf{i}_\star m \rangle ^2 \delta(\epsilon - \epsilon_m)$ (7)	$\rho_{i_\star}(\epsilon)$	$\rho_{i_\star}(\lambda)$	$\sum_m \langle \mathbf{x}_{i_\star} v_m \rangle ^2 \delta(\lambda - \lambda_m)$ (8)
$\int d\epsilon \rho_{i_\star}(\epsilon) \epsilon n(\epsilon)$ (9)	ϵ_{i_\star}	$d_{i_\star}^2$	$\int d\lambda \rho_{i_\star}(\lambda) \frac{1}{\lambda}$ (10)
$\propto \exp(-\beta \epsilon_{i_\star})$ for $\beta \rightarrow 0$ (11)	$p(\epsilon_{i_\star})$	$p(\mathbf{x}_{i_\star})$	$\propto \exp(-d_{i_\star}^2/2)$ (12)

Table 1: Quantum mechanics (QM) and machine learning (ML) formalism of sample covariance matrix and statistical distances. In order to emphasise the similitude between two approaches we adopt for statistical distances the QM bra-ket notation. The data points of the descriptor space are the ket vectors $|\mathbf{x}\rangle = \mathbf{x} \in R^{D \times 1}$ whilst the bra vectors are the transposed vectors $\langle \mathbf{x}| = \mathbf{x}^T \in R^{1 \times D}$. ρ_{i_\star} and ρ_λ are the local density of states and variance, respectively, for the state $|i_\star\rangle$ / data point $|\mathbf{x}_{i_\star}\rangle$. $p(\epsilon_{i_\star})$ is the probability of the state $|i_\star\rangle$ in the limit of high temperature, where the Fermi-Dirac distribution becomes classical Boltzmann distribution. $p(\mathbf{x}_{i_\star})$ is the marginal likelihood of the data point $|\mathbf{x}_{i_\star}\rangle$.

points $\{|\mathbf{x}_m\rangle\}$ that define the sample covariance matrix. Based on this observation, we introduce an array of statistical distances that use various weights, such as powers of eigenvalues of the sample covariance matrix, in order to approach the corresponding values from QM. For example, the QM of classical fermions (high temperature or $\beta \rightarrow 0$ suggests a weight similar to observable that gives the local energy and implies using $\lambda^\alpha \exp(\beta\lambda)$ instead of $1/\lambda$, where α and β are constants to determine. Here we propose the statistical distances with the following functional form:

$$d_{i_\star} = \left[\int d\lambda \rho_{i_\star}(\lambda) \lambda^\alpha e^{\beta\lambda} \right]^\gamma = \left[\sum_m \lambda_m^\alpha e^{\beta\lambda_m} |\langle i_\star | m \rangle|^2 \right]^\gamma \quad (13)$$

The standard MCD distance / Hotteling's T^2 estimator is given by the parameters $\alpha = -1$, $\beta = 0$, $\gamma = 0.5$. In case when the reference local energies are available, the parameters α , β , γ can be set to some optimal values. The standard choice and few sets of optimal values of these parameters for the proposed array of statistical distances are presented in the Fig.2 and Fig.3 for Fe and W, using two ML formalisms: Gaussian Approximation potential (GAP) [20] and linear ML (LML) [21, 22]. It is interesting to note that the distances inspired by the QM formalism (Fig.2b,c,d and Fig.3b,c,d) slightly outperform the standard

MCD distance / Hotteling's T^2 estimator (Fig.2a and Fig.3a) to provide a better correlation with local energies.

The perspective of using a more complex function for the weight factor can be further generalised using statistical distances defined in the framework of kernel formalism. E.g., with the procedure proposed in Ref. [23], the authors make use of the advantages of kernel whitening and kernel PCA in order to compute Mahalanobis distance in the feature space by projecting the data into the subspace spanned by the most relevant eigenvectors of the covariance matrix. This extension can entirely recover the kernel formalism that underlay the GAP potential and can potentially improve the estimation of local atomic environments via distortion score.

In conclusion, with the above considerations we found the distortion score based on various statistical distances as an appropriate measure of the distortion score of the local atomic environments. It worth to note, that in this work we gave a preference to standard MCD robust distances, which do not require any information about the energy of the system making this conjecture particularly useful and surprising. Furthermore, when this information is available, we proposed a procedure to improve this conjecture.

In order to explain the choice of MCD and statistical distances and to answer the referee's point, we have included in the revised manuscript:

- two new sections of the Methods, which synthesise the above answer. The section "*Choosing optimal outlier detection methods to provide a pertinent distortion score*" outlines the criteria that should be fulfilled by outlier detection methods in order to be valuable for the materials science community and to provide an appropriate distortion score. The second section "*Statistical distances and their Quantum Mechanics (QM)-inspired variants to set out the distortion scores*" foregrounds the interrelationship between the quantum mechanics and the statistical distances formalism. Moreover, we proposed a set of statistical distances inspired by the community of physics.
- Supplementary Materials Section I "*Comparison of MCD and OCSVM to provide a distortion score of LAE*" provides a comparison between the distortion scores provided by robust MCD and Support Vector Machine (SVM). This section contains the correlation of SVM distances with local energies from the GAP potential as well as the performance of these methods for detailed structural characterization of defects.

R3-3. I also mention this because the authors state early on: "It is worth mentioning that MCD is designed for unimodal distribution. Consequently, a careful selection of the training data should be performed to avoid multimodal distributions." Maybe there are other methods that don't have that drawback? And will this type of training data selection interfere with the usefulness of the method?

At the end of the manuscript, this returns, this time in the form: "Practically, it means that the model can be trained for detection of defects within one structural type, e.g., hcp domain within fcc structure will be identified as a defect." - does this mean that MCD will be unable to identify different defects if they occur in the data together?

The referee highlights a crucial characteristic of the present approach, namely the need of preliminary analysis of the training data. We sincerely regret the lack of clarity in the submitted version of the manuscript and acknowledge the referee for pointing this issue and giving us the opportunity to better explain this step for readers. Indeed, in the submitted manuscript we mainly focus on the systems that contain one structural defect, which is analysed with respect to the underlying bulk structure with unimodal distribution. We only briefly mention in the Discussion (page 17, 2nd paragraph) that *The outlier detection techniques are universal and perform well for the defects of different origin (i.e., the same*

Figure 4: The multimodal Bayesian Gaussian analysis [24, 25] of the fcc, hcp, $\langle 100 \rangle$ and 3D SIA training atomic environments. For better visualization, the 40-dimensional descriptor space of training data is represented in 2D using linear PCA dimensionality reduction. The subplots (a-d) illustrate the GMM analysis using from $n_g = 2$ up to $n_g = 5$ Gaussians (n_g denotes the number of Gaussians). Each point on the plot represent an atomic environment. The atomic environments belonging to the same Gaussian are depicted with the same colour. The subplot (e) shows the training configurations using the same PCA representation as for the GMM analysis (a-d), whilst the colours correspond to the structural type: the $\langle 100 \rangle$ dumbbells, 3D clusters, hcp and fcc are shown in green, blue, pink and yellow, respectively.

technique can be applied for the detection and analysis of dislocations, interstitial atoms and vacancies) (...) This strategy can be of great help for analysis of structural damage produced by displacement cascades and its evolution.

At first glance, the multimodal distribution of training data in the descriptor space can be seen as a limiting factor. Below we present a strategy that solves this concern. The strategy combines the priors of the underlying physics of the specific case and the multimodal distribution analysis (using appropriate tools, such as the Gaussian mixture model).

In order to exemplify, and push the limits of the present approach, the strategy is applied to the challenging case of a high-energy displacement cascade: a perfect crystalline system that suddenly receives, through a single atom, a huge amount of kinetic energy (in this particular case 100 keV). The primary knocked atom collides with another one, which collides with another one, etc. Through this sequence of collisions / displacements, the initial amount of the received energy very fast (within few picoseconds) is translated into creation of defects. This kind of systems commonly contain a rich *zoology* of defects, which occur at the same time. Analysis of cascade calculations often implies a challenge both due to the rich variety of defects in the cloud of structural damage and due to the large size of atomic arrays (up to few tens millions atoms).

Our strategy for analysis of such systems is based on the fact that the distortion score is not necessarily one dimensional. Several measures can be associated to each atomic environment. In this case, each component of the distortion score is related to a specific unimodal distribution of the data in the descriptor space. The choice of the distortion score dimension, as well as of the corresponding training structures

is not obvious. A systematic universal solution is not possible. The configurational space of materials is unbound. However, the physics of the system indicates some good choices of the structural classes to be used for training and the corresponding dimensionality of the distortion score.

Let's take the case of the 100 keV displacement cascade in Cu and analyze the zoology of interstitial-type defects in this system. The FCC metals are known to have small interstitial defects, like $\langle 100 \rangle$ dumbbells, alone or packed into 3D structures. The larger defects are 2D and represented by faulted (Frank) or unfaulted (prismatic, perfect) dislocation loops. The fault inside the dislocation loop has the HCP structure. Consequently, the physics of the problem suggests that the distortion score of atomic environments should be 4D and each dimension can be spanned by the robust MCD distance of the atomic environments associated to each training class: FCC, HCP, $\langle 100 \rangle$ and 3D clusters. Further, we apply a Gaussian mixture model (GMM) [24, 25] to the training data represented by the four selected structural classes. The mixture model based on Gaussian is particularly convenient for modelling continuous observations in the descriptors space, such as our training data. This data will be denoted as $\mathbf{x} \in R^D$. Consequently, the data can be written as mixture of n_g Gaussians of dimension D of mean \mathbf{m}_i and covariance matrix \mathbf{S}_i :

$$p(\mathbf{x}) = \sum_{i=1, n_g} p(\mathbf{x}|\mathbf{m}_i, \mathbf{S}_i)p_i, \quad (14)$$

where p_i is the mixture weight for component i . The optimal set parameters $\mathbf{m}_i, \mathbf{S}_i, p_i$ can be obtained using standard maximum likelihood optimization. This procedure is applied simultaneously for the four training classes that were identified as necessary for the structural analysis of the displacements cascade.

The GMM analysis, from $n_g = 2$ to $n_g = 5$ Gaussians, is presented in the Fig. 4a,b,c,d. The 40-dimensional training data is visualised in 2D after a linear PCA decomposition. The GMM analysis indicates that $n_g = 4$ accurately explains the distribution of training data. In this case, the number of Gaussians is equal to the number of *physical* classes of the reference structures. This conjecture is supported by the fact that the atomic environments of each *physical* class is generated by a Gaussian and/or Maxwell-Boltzmann distributions.

The GMM analysis employing 5 Gaussians emphasises the the presence of a bi-modal distribution within the HCP class. This comes from the fact that the training HCP structures do not represent a perfect 2-layer AB close-packed structure and that A and B layers are not strictly identical from the point of view of symmetry. This bi-modal distribution raises the question if the MCD distance associated to the unimodal HCP structure or the two individual distances should be rather associated to each Gaussian distribution of the HCP. Computing a distance from each of the HCP Gaussians will mean distinguishing between A and B layers within the HCP structure, which is not the aim of the analysis. The aim is to detect the regions with HCP lattice within the FCC lattice. Consequently, in this particular case, it is sufficient to keep a single distance associated to the HCP structure.

Thus, using the 4D distortion score, it is possible to perform a detailed analysis of the system. We are able to unambiguously localise the $\langle 100 \rangle$ self-interstitial atoms as well as 3D structures, which are difficult to identify with traditional techniques.

In order to clarify the issue pointed be the reviewer, we:

- have added a paragraph in the section "Distortion score and its correlation with local atomic energy":

Here, we exemplified the case with the single reference structure, given by the underlying bcc bulk. Each LAE can be characterised by a multi-dimensional distortion score, subsequently computed with respect to various reference structures, e.g., to different structural types of bulk or even to the structures of particular defects of interest (see the analysis of a displacement cascade in see Section II of Supplementary materials).

- have added a paragraph in the end of Methods section *Minimum Covariance Determinant*:
In order to train the model on more complex structural data with multimodal distribution of LAEs, calculations of a multidimensional distortion score can be enabled by modal decomposition of the training database. For instance, a multimodal training database \mathcal{D} can be decomposed in various unimodal sub-databases $\mathcal{D}_1 \oplus \mathcal{D}_2 \oplus \dots \oplus \mathcal{D}_n$ and a statistical distance can be computed with respect to each sub-database \mathcal{D}_i , providing thus a n -dimensional distortion score. Section II in the Supplementary materials provides an example of the training database decomposition and demonstrates the utility of multidimensional distortion score for analysis of complex structural damage produced by displacement cascades.
- have added an additional section dedicated to analysis of displacement cascades in the Supplementary materials, section II *Multidimensional distortion score for analysis of displacements cascades and detection of targeted defect type*. This section demonstrates the performance of the method for analysis complex structural damage caused by the presence of multiple defect types; the ability of the method to deal with multi-modal data sets, as well its applicability for analysis of large atomic systems. In the main text of the manuscript, we have added references to this supplementary section, where it was necessary.

R3-4. My second issue is that there are a number of parameters involved, the values of which are never justified. There's for instance the dRB threshold, which decides which atoms are identified as outliers. From what I've seen in the text, this threshold takes the values 3.9, 12, 17, 2.9. Deciding the threshold above which an outlier score is interpreted as actually indicating an outlier is a far from trivial problem in outlier detection and a number of strategies have been proposed (starting from relatively straightforward extreme values approaches to more complex methods, e.g. iterative ones). Giving guidance to the practitioner as to how to set this threshold will influence how useful/successful the use of outlier detection is.

We thank the referee for this remark. We realise that this aspect was not very clear presented in the submitted version of the paper. The term threshold in the paper is related to the stratified definition of the defects in the context via distortion score and should be clearly distinguished from the standard definition of threshold in the ML community. In the context of stratified definition of defects, the threshold has two utilities. (i) The first utility is the same as in ML community: it gives the critical limit from which a data point is considered as outlier (as the referee have pointed). Outlier threshold in this work always comes from MCD model with the procedure described in [15, 16]. The training database or the choice of atomic descriptors has the impact on the threshold provided by the model. Different thresholds in Fig.5a and Fig.5b come from different training databases. (ii) The second utility of threshold in the paper is to provide the limit of the decision function used to stratify the defect (see in Figure 1 in the manuscript for the explanation of stratification) and to provide some indications regarding the related to physics, such as gathering the atoms with relevant mean forces or pointing the most distorted atoms within the defect clusters.

To clarify the choice of thresholds in the Application 2, we provide a detailed analysis of dislocation dipole stratification in the Supplementary Materials. In Application 2 (main text of the article) we demonstrate how stratification of defects based on the distortion score of local atomic environments (LAEs) can serve for the reconstruction of migration energy barriers of defects from mean force calculations. In the context of this application, dislocations represent a particular type of defects as they create a long-range displacement field, which is not localised around the dislocation lines. The atoms labeled as outliers, based on the MCD threshold, are compactly located around the dislocation line and, in contrast to other defect types (e.g., Fig.5a,b in the main text), integrating the forces over the outlier region is not sufficient

for reconstruction of the Peierls barrier. In this case, accurate reconstruction of the barrier requires including the surrounding bulk atoms (inliers in the sense of MCD threshold) that are most impacted by the elastic displacement field produced by the dislocations.

Figure 5 illustrates stratification of the simulation cells with $\frac{1}{2}\langle 111 \rangle$ dislocation dipoles gliding in the $\{211\}$ plane in bcc Fe. The corresponding reconstruction of the Peierls barrier and atomic structures are presented in Fig. 5 c,d in the main text. As expected for the dislocations, considering only atoms-outliers with $d_{RB} > 2.9$ (the threshold provided by the MCD model indicated with red dashed line in Fig. 5) results in underestimated migration barrier (main text, Fig. 5c). In order to select the relevant bulk atoms, we consider possibilities for bulk stratification along the NEB path (Fig. 5b, d, f). Here, we mainly focus on the snapshots with the reaction coordinate $\zeta = 0$, $\zeta = 1$ (Fig. 5b), $\zeta = 0.26$, $\zeta = 0.74$ (Fig. 5d) and $\zeta = 0.5$ (Fig. 5f). These configurations correspond to the minimum- and maximum-energy points along the migration path (main text, Fig. 5c). The stratification levels B with $d_{RB} = 1.45$ and C with $d_{RB} = 1.0$ within the bulk structure (shown with dashed black lines in Fig. 5b,d,f) are selected in such a way that they separate distinct distortion score layers in all the selected snapshots.

With the above considerations that driven by the physics of the problem, there are ways to choose the appropriate distortion score in order to reconstruct the right value of a particular observable, here the Peierls barrier. By decreasing gradually the defect threshold for distortion score, more and more atoms are involved in the mean force, and the absolute value of the Peierls barriers tends to the asymptotic value, close to the real value. The right value of the defect threshold should be chosen when the Peierls barrier reaches the asymptotic limits and selects the atoms that are included in volume where the atomic forces are well defined (the core region of the QM/MM methods). Beyond the fact that we can provide the energy barrier without energy calculations, the above procedure provides information about the reliability of the atomic calculations. If the region of selected atoms is extremely large and reaches the frontiers of the core region, it means that the simulation box is not sufficiently large and the core domain should be extended.

The simulation cells from the main text of the manuscript are relatively small and therefore there is a strong elastic interaction between the dislocations. In such simulation cells, consideration of at least half of the bulk atoms is indispensable for reliable reconstruction of the migration barrier. In this particular case, integration of the forces over the region with $d_{RB} > 1.0$ (Fig. 5) is sufficient for reconstruction of the Peierls barrier with 95% accuracy (main text, Fig. 5c). For larger simulation cell with less important interaction between the cores, the proportion of the relevant bulk atoms will be drastically reduced. The example of the Peierls barrier reconstruction for bigger cells is provided in the Supplementary Materials.

In addition to that, we consider the effects of the training database and the choice of descriptors on the distortion score. It is worth emphasizing that an appropriate stratification of the bulk (Fig. 5b,d,f) is ensured by the training database of the MCD model. Bulk atoms represent inliers and their accurate description can be better obtained when using the training bulk structure from the relevant MD calculations. The training bulk structures with the noise generated by random displacements within the Gaussian distribution may not necessarily provide the results with sufficient accuracy to describe the subtle details of the displacement field produced by dislocations in the bulk structure.

It is also worth mentioning that the choice of atomic descriptors and their dimensionality has an impact on the computed distortion scores. Throughout this study, we employ bispectrum SO(4) [26]. This type of descriptors ensures a good radial and angular description of the structure, therefore they are commonly used for design of interatomic ML potentials [21, 27, 28]. We find that the accurate description of the bulk structure (sufficient for the Peierls barrier reconstruction in bcc Fe) is provided by the $j_{max} = 4.0$ or $j_{max} = 4.5$ and $R_c = 5.0$. Figure 6 compares stratification of the bulk structure in the small simulation cell with the dislocation dipole obtained using the data representation with $j_{max} = 4.0$ or $j_{max} = 4.5$. The absolute values of the robust distances obtained with different j_{max} is different, but the overall pattern for the structure stratification remains similar (Fig. 6) and the subsets of atoms classified as

Figure 5: Stratification of the small simulation cell with a dislocation dipole. The corresponding structures and the Peierls barriers are provided in Figure 5 c-d in the main text. MCD analysis is performed on the structural data represented using bispectrum SO(4) with $j_{max} = 4.0$ and $R_c = 5.0$. Each point on the plot represents an individual atom in the simulation box. Subplots (a,c,e) depict the full simulation cells with 810 atoms; subplots (b,d,f) depict the 582 atoms identified as bulk. Subplots (a,b); (c,d); (e,f) correspond to the snapshots with the reaction coordinate $\zeta = 0$, $\zeta = 1$; $\zeta = 0.26$, $\zeta = 0.74$ and $\zeta = 0.5$ along the NEB path in Figure 5c in the main text. The red dashed lines marked as A indicate the critical threshold $d_{RB} = 2.9$ identified by the MCD model. The black dashed lines at $d_{RB} = 1.45$ and $d_{RB} = 1.0$ marked as B and C in (b,d,f) indicate the possibilities for bulk stratification that these snapshots have in common.

inliers and outliers are the same.

In order to clarify the aspect raised by the referee, we have done the following changes:

- based on the analysis provided above, we have added a new section III in the Supplementary Materials, entitled “Energy profile of the screw dislocation glide in bcc iron”. In this section we consider the case of the screw dislocations in iron in various numerical conditions: box sizes, training database or descriptor choices.
- the term threshold, which might be confusing in the context of defect stratification, was replaced throughout the paper with *MCD threshold* or *SVM threshold* when the value is provided by a particular ML algorithm, and by *stratification threshold* or *defect threshold* when the value is chosen based on materials science considerations.

R3-5. This is also the case, although less problematic, for the parametrization of the SVM. The authors set γ for the kernel to 0.03, without explaining why. The value of the complexity parameter C is not mentioned. For the transferability analysis, $p=4$ and $C=0$ seem to be justified by the values in the literature but “kernel similar to that, which was originally used for the design of the ML potential” sounds as if it’s precisely not the same.

Good point from the reviewer. The kernel parameters for SVM were set exactly corresponding to those

Figure 6: Stratification of the bulk from small simulation cell with dislocation dipole (reaction coordinate $\zeta = 0$, $\zeta = 1$ in Fig. 5c in the main text) represented with bispectrum SO(4) with (a) $R_c = 5.0 \text{ \AA}$ and $j_{max} = 4.0$ and (b) $R_c = 5.0 \text{ \AA}$ and $j_{max} = 4.5$. The critical thresholds identified by the MCD model (a) $d_{RB} = 2.90$ and (b) $d_{RB} = 3.23$ are indicated with red dashed lines.

of the tested GAP potential, which is a necessary condition to check the performance of the potential. Indeed, saying “kernel similar to that” is a bad way to describe the choice of parameterization. In order to avoid confusion and to make it more clear for the readers, we have done a careful proof-reading and corrected this phrase in the last paragraph of the *Support Vector Machine* Section in Methods. The corrected sentence is :

For the transferability analysis of the GAP potential, we employ a polynomial kernel identical to that, which was originally used for the design of the ML potential [20], i.e., with $p = 4$ and $c = 0$ (homogeneous kernel).

R3-6. Language remarks - there’s a lot of small stuff that indicates that the paper should be proof-read again:

- page 3: “the materials properties” is missing a possessive apostrophe somewhere
- page 3: “Identification and characterization of defects provides” should be “provide” since it’s plural
- page 3: “ and a by rapidly growing number of fast exploring, biased in energy” - apart from the fact that I suspect it should be “based in energy”, this partial phrase doesn’t work grammatically

I’m not going to go through everything I highlighted in the text, just, I didn’t understand “The subtle elastic interaction of dislocations rises relaxation patterns that are captured by the distortion score.” at all.

We thank the referee for these linguistic remarks. In order to improve the quality of the paper, we have done a careful proof-reading of the text.

R3-7. All in all, while I found the paper interesting and convincing, the authors need to 1) decide whether they want to argue in favor of outlier detection or push MCD as the method of choice. In either case, alternatives should be discussed and/or evaluated, and the choice of MCD motivated. In the latter case, some of the too general claims, as well as the title, need to be revised. 2) explain how one would set parameters, particularly the cutoff value of the outlier score, and how to select data to arrive at a reliable inlier model. 3) improve the language.

We thank the referee one more time for the appreciation of our work and for all the constructive

remarks and suggestions that helped improving the quality and the impact of the manuscript. All the points raised by the referee were detailed in the answers above.

References

1. Desjonquères, M. C. & Spanjaard, D. *Concepts in Surface Physics* (Springer-Verlag, New York, 1993).
2. Finnis, M. W. *Interatomic forces in condensed matter* (Oxford University Press, Oxford, 2003).
3. Pettifor, D. G. New many-body potential for the bond order. *Phys. Rev. Lett.* **63**, 2480 (1989).
4. Pettifor, D. G. *Bonding and structure of molecules and solids* (Oxford University Press, Oxford, 1996).
5. Horsfield, A. P., Bratkovsky, A. M., Fearn, M., Pettifor, D. G. & Aoki, M. Bond-order potentials: theory and implementation. *Phys. Rev. B* **53**, 12694 (1996).
6. Mulliken, R. S. Electronic Population Analysis on LCAO–MO Molecular Wave Functions. I. *The Journal of Chemical Physics* **23**, 1833–1840. eprint: <https://doi.org/10.1063/1.1740588> (1955).
7. Sernicola, G. *et al.* In situ stable crack growth at the micron scale. *Nat. Commun.* **8**, 108 (2017).
8. Kermode, J. R. *et al.* Low-speed fracture instabilities in a brittle crystal. *Nature* **455**, 1224–1227 (2008).
9. Kermode, J. R. *et al.* Low speed crack propagation via kink formation and advance on the silicon (110) cleavage plane. *Phys. Rev. Lett.* **115**, 135501 (2015).
10. Schütt, K. T., Arbabzadah, F., Chmiela, S., Müller, K. R. & Tkatchenko, A. Quantum-chemical insights from deep tensor neural networks. *Nature Communications* **8**, 1–8 (2017).
11. Bartók, A. P. *et al.* Machine learning unifies the modeling of materials and molecules. *Sci Adv* **3**, e1701816 (2017).
12. Chmiela, S., Sauceda, H. E., Müller, K.-R. & Tkatchenko, A. Towards exact molecular dynamics simulations with machine-learned force fields. *Nature Communications* **9**, 1–10 (2018).
13. Noé, F., Olsson, S., Köhler, J. & Wu, H. Boltzmann generators: Sampling equilibrium states of many-body systems with deep learning. *Science* **365**, 1147 (2019).
14. Hauseux, P. *et al.* From quantum to continuum mechanics in the delamination of atomically-thin layers from substrates. *Nature Communications* **11**, 1–8 (2020).
15. Rousseeuw, P. J. & Hubert, M. Anomaly detection by robust statistics. *WIREs Data Min. Knowl.* **8**, e1236 (2018).
16. Hubert, M., Debruyne, M. & Rousseeuw, P. J. Minimum covariance determinant and extensions. *WIREs Comp. Stat.* **10**, e1421 (2018).
17. Ghasemi, E. *et al.* An evaluation of Mahalanobis-Taguchi system and neural network for multivariate pattern recognition. *J. Ind. Syst. Eng.* **1**, 139 (2007).
18. Su, C.-T., Wang, P.-C., Chen, Y.-C. & Chen, L.-F. Data mining techniques for assisting the diagnosis of pressure ulcer development in surgical patients. *J. Med. Syst.* **36**, 2387 (2012).
19. Ghasemi, E., Aghaie, A. & Cudney, E. Taguchi system: a review. *Int. J. Qual. Reliab. Manag.* **32**, 291 (2015).
20. Dragoni, D., Daff, T. D., Csányi, G. & Marzari, N. Achieving DFT accuracy with a machine-learning interatomic potential: Thermomechanics and defects in bcc ferromagnetic iron. *Phys. Rev. Materials* **2**, 013808 (2018).

21. Goryaeva, A. M., Maillet, J.-B. & Marinica, M.-C. Towards better efficiency of interatomic linear machine learning potentials. *Comput. Mater. Sci.* **166**, 200–209 (2019).
22. Goryaeva, A. M., Umm-Toc, W. & Marinica, M. C. *MiLaDy - Machine Learning Dynamics* (CEA, Saclay, 2015-2020).
23. Nader, P., Honeine, P. & Beuseroy, P. Mahalanobis-based one-class classification. *2014 IEEE International Workshop on Machine Learning for Signal Processing (MLSP)*.
24. Jordan, M. I. & Jacobs, R. A. Hierarchical mixtures of experts and the EM algorithm. *Neural Computation* **6**, 182 (1994).
25. Marin, J. M., Mengersen, K. & Robert, C. P. *Bayesian modelling and inference on mixtures of distributions* ().
26. Bartók, A. P., Kondor, R. & Csányi, G. On representing chemical environments. *Phys. Rev. B* **87**, 184115 (2013).
27. Thompson, A., Swiler, L., Trott, C., Foiles, S. & Tucker, G. Spectral neighbor analysis method for automated generation of quantum-accurate interatomic potentials. *J. Comp. Phys.* **285**, 316–330 (2015).
28. Wood, M. A. & Thompson, A. P. *Quantum-accurate molecular dynamics potential for tungsten* Preprint at <https://arxiv.org/abs/1702.07042>. 2017.

REVIEWERS' COMMENTS:

Reviewer #3 (Remarks to the Author):

The authors have responded to the issues I raised in my original review and have added text to the main manuscript and the supplementary material that addresses my concerns.

The manuscript still has quite a few language errors, many of which can be traced back to the native language of the authors, I think: "the" where there shouldn't be an article or an "a" instead, missing "the" where one would be needed, "similitude", and others.

While they don't take away from understanding the work, correcting them would make for nicer reading so if the authors have access to someone with a native-level grasp of English, some more proof-reading might be good.

Also, "can not" means that one is able to not do a thing, whereas not being able to do a thing is "cannot".

Response reviewers
Ref. No.: NCOMMS-20-05687

Title: Reinforcing materials modelling by encoding the structures of defects
in crystalline solids into distortion scores

Authors: Alexandra M. Goryaeva et al.

June 23, 2020

Reviewer #3

R3-1. The authors have responded to the issues I raised in my original review and have added text to the main manuscript and the supplementary material that addresses my concerns.

We thank again the referee for the careful reading of our manuscript and for the recognition of our work. Moreover, all her / his suggestions through the review process have definitely improved the manuscript and we express our sincere gratitude for this constructive criticism.

R3-2. The manuscript still has quite a few language errors, many of which can be traced back to the native language of the authors, I think: “the” where there shouldn’t be an article or an “a” instead, missing “the” where one would be needed, “similitude”, and others. While they don’t take away from understanding the work, correcting them would make for nicer reading so if the authors have access to someone with a native-level grasp of English, some more proof-reading might be good.

Following the suggestions of the referee, we have done our best and performed a careful proof reading of the paper in order to improve the language and reduce the number of typos.

R3-3. Also, “can not” means that one is able to not do a thing, whereas not being able to do a thing is “cannot”.

We thank again the referee for the careful reading of our paper. The two typos related to “can not” are corrected.